# Spin Injection and Transport in Organic Materials

**DOI:** 10.3390/mi10090596

**Published:** 2019-09-10

**Authors:** Qipeng Tian, Shijie Xie

**Affiliations:** 1School of Physics, Shandong University, Jinan 250100, China; 2School of Physics, State Key Laboratory of Crystal Materials, Shandong University, Jinan 250100, China

**Keywords:** organic spin valves, organic magnetic field effects, organic excited ferromagnetism, organic spin currents, organic spintronics

## Abstract

This review introduces some important spin phenomena of organic molecules and solids and their devices: Organic spin injection and transport, organic spin valves, organic magnetic field effects, organic excited ferromagnetism, organic spin currents, etc. We summarize the experimental and theoretical progress of organic spintronics in recent years and give prospects.

## 1. Introduction

In the past three decades, there have been two historic breakthroughs in the fields of spintronics: In 1988, Fert and Grunberg independently discovered the giant magnetoresistance (MR) effect in Fe/Cr multilayer films [1,2]; in 1996, Slonczewski and Berger predicted that spin torque transfer [3,4], and will soon be experimentally confirmed [5,6]. In the same year, Wolf first proposed “Spintronics” (namely spin electronics) [7], which included not only the storage of spins, but also the transport of spins. After years of development, spintronics has become an independent emerging discipline, some basic concepts and theories have been established.

Spin relaxation time $\tau_{s}$ is a commonly used microscopic physical quantity to describe spin polarization transport and can be given by $\tau_{s}^{-1}=\tau_{↑↓}^{-1}+\tau_{↓↑}^{-1}$, where $\tau_{ss^{\prime }}$ represents the average time from spin s to $s^{\prime }$. It sets the time scale for the loss of spin polarization, and then sets the spatial scale, namely the spin relaxation length $l_{s}$. The spin relaxation length is defined as the distance of the electron travels in time $\tau_{ss^{\prime }}$. The spin relaxation time of organic semiconductor at room temperature was within the range of 10^−7^–10^−5^ s, which is much larger than 10^−10^ s in metals.

Compared to conventional inorganic materials, organic materials are easier to process and synthesize. The most important is the “soft” structure that allows it to form good interfacial contact with the electrodes. Organic semiconductors have fairly weak spin-orbital coupling and hyperfine interaction, so the spin diffusion length of electrons can be very long. These properties make organic semiconductors one of the preferred materials for spintronics, which is considered an attractive topic in spintronics. Organic spintronics consists of organic functional materials that intersect with chemistry and spintronics that intersect with physics. It not only broadens our understanding of the organic world, but also has important significance in the applications of spintronics and technology.

In 2002, the Dediu group reported on spin injection and transport in organic materials for the first time [8]. In 2004, Xiong et al. prepared a Co/tris-8-hydroxyquinoline aluminum (Alq_3_)/La_1−x_Sr_x_MnO_3_ (LSMO) device, which measured 40% MR at low temperature and realized the organic spin valve effect [9]. In the same year, Francis et al. applied a small external magnetic field (≤100 mT) to organic devices without any magnetic elements, indium-tin-oxide (ITO)/poly(3,4-ethylenedioxythiophene) (PEDOT)/polyfluorene/Ca and ITO/PEDOT/sexithienyl (T_6_)/Ca/Al, they were surprised to find that more than 10% of magnetoresistors could be observed at room temperature and the MR is related to the thickness of organic layer and external bias [10,11]. Further studies show that not only organic magnetoresistance (OMAR), but also strong magnetic response in photoluminescence (PL), electroluminescence (EL) and the photocurrent (PC) in organic devices were found. Since this phenomenon is difficult to observe in inorganic devices, the organic magnetic field effect has been widely concerned by the fields of physics, chemistry, materials and electronics. Since 2009, people have also revealed the organic multi-iron [12,13], excited ferromagnetic [14,15,16] and spin current [17,18], showing the infinity of organic materials in terms of functionality.

Organic semiconductors have special carriers and a spin-charge relationship. As shown in Table 1, there are solitons, polarons, bipolarons, excitons, biexcitons and trions in organic semiconductors. All these carriers are spatial localized due to the strong electron–phonon coupling, which is different from the extended electrons or holes in normal semiconductors.

Organic materials have two special spin related interactions. One is the hyperfine interaction. The spin or magnetic moment of the hydrogen nucleus interacts with the spin of the π-electron [19]:(1)$$H_{f}=\sum_{n}a_{n}{\vec{s}}_{n}\cdot {\vec{I}}_{n},$$
where $a_{n}$ represents the hyperfine interaction intensity at lattice points (molecules), and $\vec{I_{n}}$ represents the nuclear spin. Another is the spin-orbit interaction or spin-orbit coupling. From Dirac theory we have the following form [20],
(2)$$H_{so}=\frac{1}{4m^{2}c^{2}}\frac{1}{r}\frac{dV}{dr}\left(\vec{s}\cdot \vec{l}\right)=\frac{1}{2}\xi (r)\left(\vec{s}\cdot \vec{l}\right).$$

Obviously, the spin-orbit coupling only appears for angular momentum l≠0 of the electron space state. For electrons moving around the nucleus, $V=Ze^{2}/r$, r∝Z, one gives $\xi \propto Z^{4}$, which means that heavy atoms have greater spin-orbit coupling strength. In general, organic materials are mainly composed of elements with a lower atomic number, and therefore the spin-orbit coupling of organic materials is generally considered to be weak.

Considering the spin-related interactions, the electron spin in a material is generally no longer a good quantum number, and the electron state will be a spin mixed state. In 2010, Tarafder et al. calculated the spin polarization of charged Alq_3_ with DFT. It is found that a charged Alq_3_ molecule has a spontaneous magnetization proportionally to the charge quantity [21]. Later study on thiophene ogalimars found a complex relationship between spontaneous magnetization and charge, as shown in Figure 1. Magnetization is not only related to the charge quantity, but also to the molecular size, electron–electron interaction, spin-orbit coupling and, especially, the localization of the electronic state. If the doped charges are more than one electronic charge unit, the molecular magnetic moment will decrease. In particular, when a molecule contains two electrons, it is found that the magnetic moment becomes zero [22,23].

In organic semiconductors, a charge state can be considered as a combination of polaron and bipolaron eigenstate [24],
(3)$$\psi =a_{p_{↑}}\phi_{p_{↑}}+a_{p_{↓}}\phi_{p_{↓}}+a_{bp}\phi_{bp},$$
where $\phi_{p_{↑}}$, $\phi_{p_{↓}}$ and $\phi_{bp}$ are the eigenstates of the spin up polaron, spin down polaron and the spin zero bipolaron. Then the magnetic moment of the charge state is:(4)$$m=\left({\left|a_{p_{↑}}\right|}^{2}-{\left|a_{p_{↓}}\right|}^{2}\right)/\left({\left|a_{p_{↑}}\right|}^{2}+{\left|a_{p_{↓}}\right|}^{2}\right).$$

Organic semiconductors are composed of small molecules or polymers. In a molecular crystal or a polymer chain, carrier transport can be regarded as band transport to some extent. However, transport in disordered organic materials is mainly realized by hopping. The appearance of polarons and bipolarons complicates the transport process in organic materials. These characteristics make the study of organic spintronics challenging but attractive.

## 2. Organic Spin Valve 

### 2.1. Organic Spin Valve

In 2002, Dediu first reported spin injection and transport in organic semiconductors [8]. They used semi-metallic colossal magnetoresistance (CMR) material LSMO as a polarized electron injector, and organic layer used small molecule T_6_ to prepare organic device LSMO/T_6_/LSMO. The device structure is shown in Figure 2. The direction of magnetization of the electrode is disordered before a magnetic field is applied, and their magnetization directions become parallel after the magnetic field is applied. The negative MR in experiment (shown in Figure 3) indicated that the device resistance decreases after the magnetic field is applied. With the increase of the thickness of the organic layer, the MR decays to zero, from which they estimated that the spin relaxation length in the organic layer is about 100–200 nm.

In 2004, Xiong et al. prepared LSMO/Alq_3_/Co devices using LSMO and Co with different coercivity as electrodes [9]. MR value measured at low temperatures is up to 40%, indicating that the device has a significant spin valve effect. Majumdar et al. further studied the effects of spin relaxation [25]. It has been found that polymer devices at room temperature still have 1% magnetoresistance, which is hardly observed in small molecule devices. For the device Fe/Alq_3_/Co, the magnetoresistance is 5% at a low temperature of 11K, and when the temperature is raised to 90K, the magnetoresistance is almost reduced to zero, indicating that the spin relaxation length in the polymer is more long [26]. Pramanik prepared Ni/Alq_3_/Co nanowire devices (length 50 nm, intermediate layer thickness 30 nm), they also found that after the temperature rise, the device magnetoresistance becomes small [27]. It is worth mentioning that their experiments indirectly measured that the spin relaxation time of the device is much longer than that of inorganic materials. To further increase the OMAR, Dediu et al. improved the experiment. They prepared LSMO/Alq_3_/Al_2_O_3_/Co devices, which the insulating Al_2_O_3_ layer prevents interdiffusion and reaction between the electrodes and the Alq_3_ layer so that the device has a clear interface. Then an apparent room temperature MR is observed [28].

As a special organic material, graphene has high carrier mobility, adjustable carrier concentration, weak spin-orbital coupling and spin diffusion length over micron at room temperature. Therefore, it is one of the alternative materials for long distance spin transport. MR up to 10% and spin diffusion length over 100 μm in graphene is observed. In 2006, Hill et al. first used a FeNi alloy as a ferromagnetic electrode to prepare a graphene spin valve [29]. Subsequently, multilayer graphene spin valves with Co and Fe as electrodes and graphene spin valves with Al_2_O_3_ as tunneling layers were also successfully prepared. Using the magnetic properties of graphene nanoribbon, some novel graphene spin devices have also been proposed. For example, applying a transverse electric field to the zigzag graphene nanoribbons (ZGNR) with the anti-ferromagnetic ground state to make the band gap of one spin larger and the opposite spin band gap smaller, so that ZGNR presents semi-metallic properties to be the spin injector or detector.

Carbon nanotube is also a special organic material. In 1999, Tsukagoshi et al. prepared the first multi-walled carbon nanotube spin device and 9% MR was observed at low temperature 4.2K [30]. Selecting a higher spin-polarized magnetic electrode can further increase the MR. For example, the MR of carbon tube devices prepared with La_0.7_Sr_0.3_Mn_3_ can be as high as 61%.

Graphene and carbon tubes are two very special organic materials, and their spintronic properties have been introduced in many special literatures. In this chapter, we mainly focused on small molecules or polymers.

### 2.2. Tunneling Theory of OMAR

Organic spin valve can be simply understood from the tunneling theory. Assuming that spin is conserved during electron tunneling, the probability of tunneling is proportional to the product of the state densities at the Fermi surface *D_F_* of the electrodes at both ends. Since the conductance is proportional to the probability of tunneling, the conductance of the magnetization direction parallel and antiparallel between the left and right magnetic electrodes are respectively expressed as follows,
(5)$$G_{P}\propto D_{F(L)}^{↑}D_{F(R)}^{↑}+D_{F(L)}^{↓}D_{F(R)}^{↓},$$
(6)$$G_{AP}\propto D_{F(L)}^{↑}D_{F(R)}^{↓}+D_{F(L)}^{↓}D_{F(R)}^{↑},$$
where the superscripts ↑ and ↓ represent spin polarization up and down respectively, the subscripts L and R represent Left and right magnetic electrode respectively.

Since the resistance R∝1/G, the tunneling MR can be calculated as follows [9],
(7)$$MR=\frac{R_{AP}-R_{P}}{R_{p}}=\frac{2P_{1}P_{2}}{1-P_{1}P_{2}},$$
where $R_{P},R_{AP}$ are the device resistance when the magnetization direction of the electrodes at both ends is parallel and antiparallel, respectively, and *P*_1_ and *P*_2_ are the polarization of the electrodes. In an actual device LSMO/Alq_3_/Co, Co atoms may penetrate into the organic layer to reduce the effective thickness of the organic layer, which can be corrected to the polarization reduction at the Co-end interface, ${P^{\prime }}_{2}=P_{2}e^{-\left(d-d_{0}\right)/\lambda_{s}}$, where *d* is the thickness of the organic layer, *d*_0_ the thickness of the Co permeable layer and $\lambda_{s}$ the spin diffusion length of the organic layer. Then the MR can be obtained as follows [9],
(8)$$TMR=\frac{R_{AP}-R_{P}}{R_{p}}=\frac{2P_{1}P_{2}e^{-\left(d-d_{0}\right)/\lambda_{s}}}{1-P_{1}P_{2}e^{-\left(d-d_{0}\right)/\lambda_{s}}}.$$

Take $P_{1}P_{2}=0.32$, $d_{0}=87\;nm$ and $\lambda_{s}=45\;nm$, theoretical curves are in good agreement with experimental data [9], as shown in Figure 4.

### 2.3. Organic Spin Injection and Transport Theory

If we considered the transport process within the organic layer, a simple “two-fluid” model could be applied [31]. It is suitable that the spin scattering length is much smaller than the electron scattering length. In organic devices, the injected electrons will form polarons and bipolarons. Since a bipolaron has no spin, we had to generalize the “two-fluid” model to a “three-fluid” one: Spin-up polarons, spin-down polaron and spinless bipolaron. 

The current spin polarization is usually described as $\alpha =\left(j_{↑}-j_{↓}\right)/\left(j_{↑}+j_{↓}\right)$ with the current density $j_{s}$ with spin s. However, in an organic situation it should be modified as $\alpha =(j_{p↑}-j_{p↓})/(j_{p↑}+j_{p↓}+j_{bp})$, where $j_{p↑},j_{p↓}$ and $j_{bp}$ are the current density contributed by spin up polarons, spin down polarons and spinless bipolarons.

From the spin diffusion equation and Ohm’s law, the current spin polarization at the interface can be obtained as follows [32,33]:(9)$$\begin{array}{l} \alpha_{0}=\gamma \beta_{0}\frac{\sigma }{\sigma_{FM}}\cdot \frac{\lambda_{FM}}{\lambda_{p}}\cdot \frac{1+\frac{1}{4\beta_{0}}\cdot \frac{\sigma_{FM}}{\lambda_{FM}}\left(\frac{1}{G_{↓}}-\frac{1}{G_{↑}}\right)\left(1-\beta_{0}^{2}\right)}{\left(\gamma \frac{\sigma }{\sigma_{FM}}\cdot \frac{\lambda_{FM}}{\lambda_{p}}+1\right)-\beta_{0}^{2}+\frac{\gamma }{4}\cdot \frac{\sigma }{\lambda_{p}}\left(\frac{1}{G_{↓}}+\frac{1}{G_{↑}}\right)\left(1-\beta_{0}^{2}\right)} \end{array},$$
where the polaron ratio is $\gamma =n_{p}/(n_{p}+n_{bp})$. Obviously, when γ=0, current spin polarization will also be zero as the carriers are all bipolarons without spin. The maximum current polarization occurs at γ=1, where the carriers are all polarons with spin. In addition, we found that the presence of polarons result in a significant current spin polarization. For example, when the polaron ratio was only 20%, the current spin polarization value was 90% of that when the carriers were all polarons. Therefore, polarons are the effective spin carriers of the spin polarized current. Even if a small number of polarons are present, large current spin polarization can be obtained in organic semiconductors. From Equation (9) we see that the spin polarization is proportional to $\sigma /\sigma_{FM}$, so a large spin polarization or injection can be obtained when the conductance of the interlayer matches that of the contact, i.e., $\sigma =\sigma_{FM}$.

In the actual transport process, polarons and bipolarons do not exist independently. They may transform each other, which can be described by the following set of equations [34].
(10)$$dn_{↑(↓)}/dt=-kn_{↑}n_{↓}+bN,$$
(11)$$dN/dt=kn_{↑}n_{↓}-bN,$$
where $n_{↑\left(↓\right)}$ is the concentration of the spin-up (spin-down) polarons, and *N* is the concentration of the bipolarons. The spin polarization is defined as,
(12)$$P=(n_{↑}-n_{↓})/(n_{↑}+n_{↓}).$$

Due to the existence of bipolarons in organic semiconductors, this definition has different meanings compared to that in traditional inorganic semiconductors as bipolarons do not contribute to spin polarization. However, its existence will affect the performance of organic spin devices.

Based on Landau-Büttiker theory and quantum dynamics, we could give a microscopic understanding of organic spin injection and transport. In 2002, Zwolak et al. explored the spin-related transport phenomena of ferromagnetic/DNA/ferromagnetic sandwich structures [35]. After calculating the MR of the device by the Green’s function method, it was found that, if Fe and Ni were used as the ferromagnetic layer of the device, the MR effects of 16% and 26% were obtained at room temperature, respectively. Based on the NEGF-DFT theory, the MR of the Ni/octanethiol/Ni tunnel-junction was up to 33%, while that of Ni-octane-Ni was up to more than 100%. Microscopic calculations are mainly for a molecular device and only reflect the nature of macroscopic organic devices to a certain extent. It is difficult to consider the strong electron-lattice interaction effect in organic molecules in the NEGF-DFT theoretical framework. One possible improvement is to combine DFT phonon calculations with appropriate models [36].

Adjusting the spin polarization of the interface can significantly improve the MR. Assuming that the coupling between the ferromagnetic electrode and the organic molecule (or the integral of the interface transition) is related to the spin $t_{\sigma }$, the total conductance in the parallel state is $G_{P}\propto t_{↑}t_{↑}+t_{↓}t_{↓}$, while the total conductance in the anti-parallel is $G_{AP}\propto t_{↑}t_{↓}+t_{↓}t_{↑}$. Therefore, MR can be expressed as:(13)$$MR=\frac{G_{P}-G_{AP}}{G_{AP}}=\frac{{\left(1-\gamma \right)}^{2}}{2\gamma },$$
where $\gamma =t_{↓}/t_{↑}$. Although the model is simple, it gives an inspiration on how to obtain high MR.

Due to the spin dependent scattering, spin of a carrier will no longer be conserved during transport. The reason can be attributed to spin-orbit coupling, hyperfine interactions, electron–electron interactions and other spin-related interactions. The hyperfine interactions in organic molecules are mainly derived from hydrogen nuclear spins. Nguyen et al. replaced the hydrogen atom in poly (2,5-di-octyloxy) p-phenylene ethylene (H-DOO-PPV) with a deuterium atom (D-DOO-PPV), and the experimental results showed that the MR of the spin valve based on them differs by more than one order of magnitude [32]. It indicates that the hyperfine interaction plays an important role in the spin relaxation of organic materials. However, the theoretical calculation of spin relaxation in Alq_3_ based on the spin-orbital interaction shows that the spin diffusion behavior in Alq_3_ is completely consistent with that measured by the LE-μSR technique [37], which seems to indicate that the spin-orbital coupling is the main reason for the spin relaxation in organic materials. Therefore, the role of hyperfine interactions and spin-orbit coupling in organic materials remains a problem that needs further study.

In 2003, we studied the ground state properties of CMR/organic semiconductor systems [38]. In the calculation, polyacetylene or polythiophene is taken as the organic layer, CMR perovskite material LSMO is taken as the spin injection layer. The tight-binding Hamiltonian is used to describe the interface coupling and electronic transition process of the system, especially as the model contains strong electron–phonon interactions inherent in the organic layer. By adjusting the relative chemical potential of the two materials, electrons can be transferred from the CMR to the organic layer. Micro dynamic studies show that the injected charge forms a wave packet in the organic layer, and each wave packet can contain any value of charge 0–2e and carry a certain spin. Due to the spin dependent interaction, the carrier spin is not in the eigenstate [39]. Figure 5 shows the dynamic behavior of the injected charge. After 200 fs of injection, the wave packet has formed and moves in the organic layer driven by the external electric field. The spin polarization exists in the organic layer. However, the spin polarization mainly appears near the interface, and attenuates until it disappears when it penetrates into the organic layer. The figure also shows that the injected charge still has obvious spin polarization after 1000 fs.

Applying a voltage in the vertical transport direction can adjust the effective Rashba spin-orbit coupling strength. In the quasi-one-dimensional tight-binding approach, the spin-orbit coupling can be written as follows [20]:(14)$$H_{so}=-t_{so}\sum_{n}\left[C_{n+1,↑}^{+}C_{n,↓}-C_{n+1,↓}^{+}C_{n,↑}+C_{n,↓}^{+}C_{n+1,↑}-C_{n,↑}^{+}C_{n+1,↓}\right],$$
where *t*_so_ is the coupling strength and $C_{n,s}^{+}$ ($C_{n,s}^{}$) is the creation (annihilation) operator of *π*-electrons. When a polaron moves along the molecular chain, its spin will process at the same time, as shown in Figure 6a, where the spin at site *n* is defined as $s_{z}(n,t)=\left[\rho_{↑}(n,t)-\rho_{↓}(n,t)\right]/2$ and total spin $s_{z}(t)=\sum_{n}s_{z}(n,t)$. At the initial moment, the spin of the polaron is up, and as the polaron moves, its spin gradually becomes zero and then becomes down. Figure 6b shows the evolution of $s_{z}(t)$ with the position of the polaron center [33]. It can be clearly seen that during the movement of the polaron, its spin evolves with time due to the spin-orbit coupling.

Based on the study of spin valves and spin field effect transistors, we could design a spin diode, which means that the electron spin transport in the external field is asymmetric [40]. We defined charge current as the sum of the currents of the two spin orientations, $I_{q}=I_{↑}+I_{↓}$, and spin current as $I_{s}=\left(\hbar /2e\right)\left(I_{↑}-I_{↓}\right)$. Figure 7 shows three spin current diodes. Figure 7a is a symmetric spin current, Figure 7b is a parallel spin rectification and Figure 7c is an anti-parallel spin rectification.

A molecular spin diode, such as thiophene/poly (1,4-bis(2,2,6,6-tetramethyl-4-piperidyl-1-oxyl)-butadiin (poly-BIPO), is asymmetrical in the spin space [41]. With spin related Landau-Büttiker theory, the current passing through the device and its spin polarization can be calculated [32,42]. It is found that the device exhibits a charge as well as a spin rectification.

Currently, most studies on spin transport are carried out with a molecule-based spin valve, which shows a “ferromagnetic/non-magnetic/ferromagnetic” sandwich structure and the non-magnetic interlayer is usually a thin-film molecular layer. In such a device, the spin transport process has been demonstrated to have a close correlation with spin relaxation time and charge carrier mobility of π-conjugated molecules. The spin relaxation time $\tau_{s}$ is subsequently computed using the relationship $\tau_{s}=\lambda^{2}/\left(D_{hop}+D_{ex}\right)$, where $\lambda^{2}$, $D_{hop}$ and $D_{ex}$, are the spin relaxation length $\lambda_{s}$ of molecules, the spin diffusion constants based on the hopping spin transport mode and exchange coupling mode, respectively. When the molecular layer have a high carrier concentration, generally induced via the impurity band [43,44], an exchange coupling model will provide an extra-fast speed compared to hopping transport since the spin transport process can decouple with charge transport. It appears to improve the spin transport distance. In addition, single crystal and organic cocrystals have good application prospects in high performance spin transport due to the ordered stacked aggregation with high mobility and weak spin scattering factors [45,46,47]. Despite the limitations of technologies and the lack of theoretical models, spintronic devices based on single crystals or cocrystals still attracts people to explore its rich possibilities [48].

## 3. Organic Magnetic Field Effect

### 3.1. Organic Magnetic Field Effect Experiment

For an organic device with no any magnetic elements included, at room temperature, the optoelectronic properties will respond significantly to very small magnetic fields (on the order of mT) [10,49,50]. For most of the organic devices, the MR follows empirical Lorentz $B^{2}/(B^{2}+B_{0}^{2})$ or non-Lorentz ${\left[B/(\left|B\right|+B_{0})\right]}^{2}$ [11,51]. Figure 8 shows some experimental results and fitting curves. Individual devices may follow power-law distributions such as $B^{n}$, $f_{1}/B^{2}+f_{2}/B^{4}$ or $d_{1}B^{2}+d_{2}B^{4}$ [52,53].

Figure 9 shows a typical device for experimentally studying the magnetic field effect. The metal electrodes on both sides and the intermediate organic layer form a sandwich structure. After an electric voltage is applied between the two electrodes, carriers are injected from the electrodes into the organic layer and they transport into the organic layer. Then a magnetic field is applied to study the effect of the magnetic field on the transport. 

In 1996, Frankevich et al. experimentally found that the photocurrent in the polymer poly(2,5-diheptyloxy-p-phenylene vinylene) (HO-PPV) device increases after applying a magnetic field [54]. In 2003, Kalinowski et al. studied light-emitting device of small molecules Alq_3_, and they found that the light emitting efficiency of the device is increased by 5% as the magnetic field from 0 to 300mT [55]. Subsequently, Mermer et al. discovered MR in polymer poly(9,9-dioctylfluorenyl-2,7-diyl) (PFO) devices. At room temperature, the MR of the device reaches 10% at 100mT [31,56]. Figure 10 shows the MR curve in the ITO/PFO (≈ 60 nm)/Ca device at 200 K, and it found that the MR was negative at low voltage, but it becomes a positive value at high voltage.

In 2009, Hu et al. explored the photoluminescence efficiency of organic light diodes (OLEDs) and found that pure organic N,N’-diphenyl-N,N’-bis(3-methylphenyl)-[1,1’-biphenyl]-4,4’-diamine (TPD) and 2,5-bis(5-tert-butyl-2-benzoxazolyl)-thiophene (BBOT) have no magnetic field effect, as shown in Figure 11. However, the photoluminescence efficiency of the hybrid device (doped with different ratios of poly(methyl methacrylate) (PMMA)) is improved, and the magnetic field effects produced by the different ratios of the mixture devices are also different [49].

Nguyen et al. compared the magnetic field effect of an OLED composed of a π-conjugated polymer poly(dioctyloxy)phenylenevinylene (DOOPPV) with hydrogen and deuterium (the latter having a weak hyperfine interaction) [32]. They found that devices composed of deuterated polymers showed significant narrowing of the electroluminescence magnetic field effect, as shown in Figure 12, which indicates that hyperfine interaction is one of the important factors of the organic magnetic field effect.

### 3.2. Organic Magnetic Field Effect Theory

Based on the basic current density formula, current density J depends not only on the carrier concentration *n* but also on the carrier mobility or velocity v through J=nev. For a magnetic conductance (inverse of MR) that is not very large, we have:(15)$$MC≅\frac{n(B)-n(0)}{n(0)}+\frac{v(B)-v(0)}{v(0)}=MC(n,B)+MC(v,B),$$
where MC(n,B) and MC(v,B) are the responses of concentration and mobility to the magnetic field, respectively [34]. Experimentally, it seems to reveal that both the carrier concentration and mobility respond to the magnetic field. For example, Nguyen et al. measured the dependence of the concentration of singlet excitons, triplet excitons and polarons on the magnetic field by using electroluminescence spectroscopy and charge-induced absorption spectroscopy techniques [57]. They found that all concentrations increase with the field. However, Veeraraghavan et al. measured MR in PFO, and found the magnetic field effect is related to the carrier mobility instead of the carrier concentration [58]. In addition, Ding et al. studied OLEDs mixed by N, N’-bis(l-naphthyl)-N, N’-diphenyl-1,1’-biphentl-4,4’-diamine (NPB):Alq_3_ and found that the electroluminescence magnetic field effect has a close relationship with carrier mobility [59].

The process of understanding the effects of organic magnetic fields is multi-view and gradually deepened, and many mechanisms or models are proposed. Since the magnetic field has an effect on either the spin or the orbit of π-electrons, there are spin magnetic resistance (spin-MR) and orbital magnetic resistance (orbital-MR). The following are some of the main OMAR mechanisms.

Polaron pairing mechanism: In a bipolar device, holes injected from the anode and electrons injected from the cathode form positive and negative polarons in the organic layer, respectively. They are bound together by coulomb attraction to form a pair of polarons or excitons, which can be classified into a singlet and triplet depending on the spin configuration. The change of the singlet/triplet ratio due to the magnetic field in the polaron pairs will change the radiation or current [60]. Figure 13 shows the structural changes of the single and triplet levels with different distances. *S* denotes a singlet exciton, *T* denotes a triplet exciton and *PP*^1^ is a single-state polaron pair, *PP*^3^ is a triplet polaron pair. For the polaron pairs, since the distance between the positive and negative polaron is large, there is a small spin exchange interaction energy between them, and the energy level between the triplet polaron pairs are degenerate. However, for excitons, the difference between singlet and triplet energy levels is large. When the magnetic field *B* = 0, as shown in Figure 13a, the energy level of *PP*^3^ is a triple degeneracy state, so that the singlet and the triplet are most easily converted into each other. When the magnetic field is not zero, *PP*^3^ appears to be Zeeman splitting, and thereby the triply degenerate is released, so in this case the interconversion only occurs in *PP*^1^ and $PP_{0}^{3}$ [61], as shown in Figure 13b. In a word, the mutual transformation of the single and triplet state changes with the external magnetic field, and the ratio of the singlet polaron pairs changes, so the effects of the magnetic field on the current and the like are revealed.

As an example, let us consider a pair of polarons with state ${|↑,⇑\rangle }_{1}{|↑,⇓\rangle }_{2}$. |↑,⇓〉 means the nuclei spin up and the polaron spin down. The first polaron is in eigenstate and the second one will evolve with time. Considering the effects of external magnetic fields and hyperfine fields,

(16)$$H=\omega_{0}s_{z}+\frac{a}{\hbar }\vec{s}\cdot \vec{I}.$$

The first term represents the Zeeman energy produced by the external magnetic field, $\omega_{0}=g\mu_{B}B/\hbar$, where $\mu_{B}$ is the Bohr magneton and g is the Landau factor. The second term represents the interaction of spin with the hydrogen nuclear spin $\hat{\vec{I}}$ (assumed to be ℏ/2), and a is the interaction strength. We can get the expected value of the total spin,
(17)$$s_{z}^{1+2}(t)=\frac{1}{2}\left(1+\frac{\omega_{0}^{2}}{\omega_{0}^{2}+a^{2}}+\frac{\omega_{0}^{2}}{\omega_{0}^{2}+a^{2}}\cos\sqrt{\omega_{0}^{2}+a^{2}}t\right)=\frac{1}{2}+\frac{1}{2}\left(p_{P}-p_{AP}\right),$$
where $p_{P}$ and $p_{AP}$ respectively indicates the probability that the pair of polarons are in parallel and anti-parallel states. The above formula shows that the total spin of a pair of positive and negative polarons oscillates with time. The experiment found that the oscillation frequency is much higher than the recombination rate of their formation (bound state), so we take the average and get the recombination rate,
(18)$$\gamma ={\bar{p}}_{P}\gamma_{P}+{\bar{p}}_{AP}\gamma_{AP}=\left(\frac{1}{2}+\frac{\omega_{0}^{2}}{2\left(\omega_{0}^{2}+a^{2}\right)}\right)\gamma_{P}+\frac{\omega_{0}^{2}}{2\left(\omega_{0}^{2}+a^{2}\right)}\gamma_{AP},$$
where $\gamma_{P}$ and $\gamma_{AP}$ respectively indicate the rate of recombination for parallel and antiparallel pairs. Therefore,
(19)$$\frac{\Delta \gamma }{\gamma }=\frac{\gamma (B)-\gamma (B=0)}{\gamma (B=0)}=\beta \frac{\omega_{0}^{2}}{\omega_{0}^{2}+a^{2}},$$
where the parameter $\beta =\frac{\gamma_{P}-\gamma_{AP}}{\gamma_{P}+\gamma_{AP}}$. The above formula represents the Lorentz type MR found experimentally.

Bipolaron mechanism: In a unipolar device, there is only one kind of charge (electron or hole) injected. In this case, a bipolaron model is proposed, which is based on the transformation between the polaron and the bipolaron. Using diffusion theory we write the transformation equation as [34]:(20)$$\{\begin{array}{l} \frac{dn_{pp}}{dt}=-\gamma_{ap}n_{pp}+{\gamma^{\prime }}_{ap}n_{ap}\\ \frac{dn_{ap}}{dt}=\gamma_{ap}n_{pp}-{\gamma^{\prime }}_{ap}n_{ap}+kn_{bp}-bn_{ap}\\ \frac{dn_{bp}}{dt}=-kn_{bp}+bn_{ap} \end{array},$$
where $n_{pp}(n_{ap})$ is the concentration of spin-parallel (anti-parallel) polaron pairs and $n_{bp}$ is the concentration of the bipolarons. $-\gamma_{ap}n_{pp}$($-{\gamma^{\prime }}_{ap}n_{ap}$) means the reduction of the spin-parallel polaron pairs (bipolarons) due to the external magnetic field and the hyperfine field. Parameter *b* represents the recombination rate of polaron pair to form bipolaron. *k* represents the dissociation rate of bipolaron; $\gamma_{ap}$ represents the conversion rate of spin parallel polaron pair to spin antiparallel polaron pair and ${\gamma^{\prime }}_{ap}$ is the opposite process above. Under the external magnetic field and the hyperfine effect, the above conversion rate can be obtained as follows,
(21)$$\gamma_{ap}=\frac{1}{2}[1-\frac{\omega^{4}}{8{\left(\omega^{2}+\alpha^{2}\right)}^{2}}]\gamma_{0},$$
(22)$$\gamma_{ap}^{\prime }=\frac{1}{2}[1+\frac{\omega^{4}}{8{\left(\omega^{2}+\alpha^{2}\right)}^{2}}]\gamma_{0},$$
where $\gamma_{0}$ is a constant. Combining current density:(23)$$J\left(B\right)=2e\left[n_{pp}\left(B\right)+n_{ap}\left(B\right)\right]\nu_{p}+2en_{bp}\left(B\right)\nu_{bp},$$
the magnetic conductance is given as:(24)$$\mathrm{MC}=MC_{\infty }\frac{\omega^{4}}{\omega^{4}+2\beta a^{2}\omega^{2}+\beta a^{4}},$$
where $MC_{\infty }=\frac{2(1-\alpha )k/b}{\left(\alpha +2k/b)(7+16k/b\right)}$ is the value of the saturated MC. $\alpha =v_{bp}/v_{p}$ is the rate ratio of bipolarons and polarons, and $\beta =1+\frac{1}{16k/b+7}$. Since β≈1, when the external magnetic field is much larger than the hyperfine equivalent field $B\gg B_{\mathrm{hf}}$(ω≫a), $\mathrm{MC}\approx MC_{\infty }\frac{\omega^{2}}{\omega^{2}+2\beta a^{2}}=MC_{\infty }\frac{B^{2}}{B^{2}+B_{\mathrm{hf}}^{2}}$ and this form is the empirical formula of Lorentz form. When the external magnetic field is small, i.e., $B<B_{\mathrm{hf}}$, the MC curve deviates from the Lorentz curve, as shown in the inset in Figure 14. In organic semiconductors, the concentration of polarons is usually greater than that of bipolarons, so a positive MC is obtained. With the appropriate parameters, a calculation result equivalent to 2% of the experimental value can be obtained [34]. It is worth noting that the calculated MC saturation value has no relationship with the hyperfine equivalent field, as shown in Figure 14. These calculation results are completely consistent with the experimental measurement [36].

Exciton quenching model: The long lifetime of a triplet exciton makes it easy for a polaron to collide with it, which leads to two consequences: One is to obstruct the movement of the polaron; the other is to react to each other. One of the reaction channels is to form a short-lived singlet exciton [62],

(25)$$P^{-\sigma }+T_{ex}^{\sigma \sigma }\to P^{\sigma }+S_{ex}^{0}.$$

Two triplet excitons also annihilate into singlet excitons [63],

(26)$$T_{ex}^{\sigma \sigma }+T_{ex}^{-\sigma -\sigma }\to S_{ex}^{0}.$$

A magnetic field will change the collision probability, which affects the mobility of polarons, so that the current of the device is changed by the magnetic field. Through non-adiabatic kinetic evolution methods, Sun et al. studied the scattering process between a negative polaron and a triplet exciton on the polyacetylene molecular chain, as shown in Figure 15 [64]. The figure shows that the transport of a polaron is indeed hindered by a triplet exciton.

The OMAR can also be understood based on the spin blocking effect. Using the tight-binding model of organic small molecule crystals, the Hamiltonian of the system is [65]:(27)$$H_{TO}=\sum_{n,s}\left[-\tau +\alpha (u_{n+1}-u_{n})](C_{n+1,s}^{+}C_{n,s}+C_{n,s}^{+}C_{n+1,s}\right)+\sum_{n}\frac{1}{2}M{\dot{u}}_{n}^{2}+\sum_{n}\frac{1}{2}K{\left(u_{n+1}-u_{n}\right)}^{2},$$
where $u_{n}$ is the deviation of the *n*th molecule from the equidistant molecular arrangement, τ is the electron transfer integral before the molecular deviation and α is an electron–phonon coupling constant. M is the molecular mass and K is the intermolecular elastic coupling constant. The external magnetic field and hyperfine interaction are given by equation (16).

Driven by an external electric field, the evolution of the electronic state is given by the time-dependent Schrödinger equation,

(28)$$\begin{array}{c} i\hbar \frac{\partial }{\partial t}Z_{\mu ,n,\sigma }(t)=-[\tau -\alpha (u_{n}-u_{n-1})]Z_{\mu ,n-1,\sigma }(t)-[\tau -\alpha (u_{n+1}-u_{n})]Z_{\mu ,n+1,\sigma }(t)\\ +g\mu_{B}\sigma \left(B+B_{hyp,n}\right)Z_{\mu ,n,\sigma }(t)-eE(na+u_{n})Z_{\mu ,n,\sigma }(t) \end{array}$$

At the same time, the atomic nuclei (or CH units) will evaluate with the electronic states, which are determined by the classical Newtonian equation of motion:(29)$$m{\ddot{u}}_{n}(t)=K(u_{n+1}+u_{n-1}-2u_{n})+2\alpha [\rho_{n,n+1}(t)-\rho_{n-1,n}(t)]-eE[\rho_{n,n}(t)-1],$$
where the electronic state and the density matrix, respectively, are:(30)$$\psi_{\mu }=\left(\begin{array}{l} \varphi_{\mu ↑}\\ \varphi_{\mu ↓} \end{array}\right)=\left(\begin{array}{l} \sum_{n}Z_{\mu ,n,↑}|n\rangle \\ \sum_{n}Z_{\mu ,n,↓}|n\rangle \end{array}\right),$$

(31)$$\rho_{m,n}=\sum_{\mu }Z_{\mu ,m,\sigma }^{*}f_{\mu }Z_{\mu ,n,\sigma }.$$

Equations (28) and (29) can be solved by the Runge–Kutta method with 8-order controllable step size. At the beginning, supposing there is a polaron in the system, its velocity is solved by the above equations, which is dependent upon the magnetic field *B*, thus giving the MR of the velocity response. As shown in Figure 16, MR decreases rapidly with the decrease of hyperfine field intensity. For example, the fixed external magnetic field *B* = 80 mT, when the hyperfine field decreased from 4.6 mT to 2.4 mT, MR decreased from 1.78% to 0.88%. The OMAR effect disappears if there is no hyperfine field. 

For most OLEDs, the organic layers are amorphous, the coherent transport mechanism described before is not suitable. In this case, polarons move from one molecule to another by a hopping manner. It involves spin-conserved hopping as well as spin-reversed one, as shown in Figure 17.

To describe the hopping process, we assumed that the spin and charge hopping rates were independent, $W_{is,js^{\prime }}=\alpha_{ss\prime }\omega_{ij}$, where $\omega_{ij}$ is the normal charge hopping given by the Marcus formula:(32)ωij=tij2ℏ[πkBTλ]12exp[−(λ+εj−εi)24kBTλ],
where $t_{ij}=t_{0}\exp(-2\gamma R_{ij})$ is the transition integral between the molecules *i* and *j*. $R_{ij}=\left|R_{j}-R_{i}\right|$ is the distance between the two molecules, γ is the local factor of the wave function, which reflects the localization of the polaron and $\varepsilon_{i}$ is the on-site energy. After applying a driving electric field, the exponential factor in the equation (32) should add an energy contribution of the electric field $-eER_{ij,x}$ (*e* is the charge value of the carriers). λ is the molecular reforming energy that reflects the properties of organic materials. $\alpha_{ss}$ reflects the spin hopping rate. When we consider the hopping between molecule i and j, its spin will be affected by the external magnetic field and the hyperfine field. Hamiltonian is written as:(33)$${\hat{H}}_{ij}=g\mu_{B}\vec{s}\cdot \vec{B}+a_{i}\vec{s}\cdot {\hat{\vec{I}}}_{i}+a_{j}\vec{s}\cdot {\hat{\vec{I}}}_{j}.$$

Considering all possible spin configurations and taking the statistical average of these configurations, the spin hopping rate is obtained as [68]:(34)$$\alpha_{ss^{\prime }}=\{\begin{array}{c} \frac{B^{2}+7a_{H}^{2}/4}{B^{2}+9a_{H}^{2}/4}\;(s=s^{\prime })\\ \frac{a_{H}^{2}/2}{B^{2}+9a_{H}^{2}/4}\;(s\neq s^{\prime }) \end{array}.$$

In order to calculate the magnetic conductance in an organic device, the master equation is employed by including a spin index as [68],
(35)$$\frac{dP_{is}}{dt}=\sum_{j\neq i,s^{\prime }}\left[-W_{is,js^{\prime }}P_{is}(1-P_{js^{\prime }}\right)+W_{js^{\prime },is}P_{js^{\prime }}(1-P_{is})],$$
where $P_{is}$ is the number of occupied polarons of spin *s* of lattice *i*. The mobility of the system is expressed as:(36)$$\mu =\frac{1}{PE}\sum_{is,js^{\prime }}W_{is,js^{\prime }}P_{is}(1-P_{js^{\prime }})R_{ij,x}.$$

Then the magnetic conductance of the system is given by [34],

(37)$$\mathrm{MC}=\frac{\mu (B)-\mu (0)}{\mu (0)}.$$

With proper parameters, the calculated mobility in the absence of external magnetic field is $\mu \left(0\right)=2.54\times 10^{-5}{\mathrm{cm}}^{2}V^{-1}s^{-1}$ referring to that of Alq_3_. When an external magnetic field was applied, it was found that the mobility was significantly changed, and a magnetic conductance value of up to 75% was obtained. Figure 18 shows the change of the magnetic conductance with a magnetic field calculated at a different polaron density.

Considering the Lorentz effect of the magnetic field on moving polarons, orbital MR is addressed [70]. This effect may be significant in organic layers with asymmetric molecules or configurations. In the presence of a magnetic field, due to the charge Lorentz effect, the transition integral between two molecules can be modified to,
(38)$$t=\frac{\hbar^{2}}{m}\int [(\psi_{1}^{*}\frac{\partial \psi_{2}}{\partial x}-\frac{\partial \psi_{1}^{*}}{\partial x}\psi_{2})-\frac{2i}{\phi_{0}}A_{x}\psi_{1}^{*}\psi_{2}]|{}_{x=d/2}dydz,$$
where $A_{x}$ is the *x* component of the magnetic vector potential, $\phi_{0}=c\hbar /e$ is the magnetic flux quantum number, $\psi_{1}$ and $\psi_{2}$ are electronic states on two molecules, and they are usually local states of polarons. The integral range is the entire plane of x=d/2, located between the two molecules. If two molecules are not symmetric with respect to y=0 or z=0, the imaginary part of this integral will be obvious.

Considering the transport process of a positive polaron and a negative polaron injected into the organic layer. They meet and collide with each other. One of the yields of the collision is an exciton. The Lorentz effect generated by the magnetic field will affect the polaron motion, and then change the yield of excitons. The calculated magnetic conductance is shown in Figure 19, which is in good agreement with the experimental data. A further study also finds that the magnetic conductivity is sensitive to the structural symmetry and electron-lattice coupling strength of organic molecules, which also explains why the organic magnetic field effect (OMFE) is more significant than that in inorganic materials [71].

## 4. Organic Excited Ferromagnetism

Multiferroic materials refer to having two or more kinds of properties of ferromagnetic, ferroelectric and ferroelastic. However, due to the weak magnetoelectric coupling strength, research in this field has been slow. Since the discovery of multiferroic material BiMnO_3_ [74] and BiFeO_3_ [75] in 2003, multiferroic materials have rapidly become a research hotspot in the field of physics and material science due to their rich physical properties and potential applications in functional devices. In multiferroic materials, the charge, spin, orbital and phonons in the lattice are strongly coupled to each other, making the multiferroic material rich in physical properties, facilitating the rapid development of condensed matter physics and materials science. In addition to inorganic materials, the discovery of organic multiferroic materials has begun to attract more and more attention, providing a new idea for the application of magnetoelectric coupling multiferroic devices [76,77]. The characteristics of room temperature ferroelectric, ferromagnetic and magnetoelectric coupling were found in the organic charge transfer complex. The supramolecular design technique can assemble the electron donor and acceptor molecules into an ordered charge transfer network to realize ferroelectricity.

According to Landau’s theory, the free energy of multiferroic materials can be written as:(39)$$\begin{array}{cc} f(\vec{E},\vec{H}) & =f_{0}-P_{0i}E_{i}-M_{0i}H_{i}-\frac{1}{2}\varepsilon_{ij}E_{i}E_{j}-\frac{1}{2}\mu_{ij}H_{i}H_{j}\\ & -\alpha_{ij}E_{i}H_{j}-\frac{1}{2}\beta_{ijk}E_{i}H_{j}H_{k}-\frac{1}{2}\gamma_{ijk}H_{i}E_{j}E_{k}-... \end{array}$$
where $E_{i}$ and $H_{i}$ are electric field component and magnetic field component, $P_{0i}$ and $M_{0i}$ are spontaneous electric polarization intensity and spontaneous magnetization intensity and $\varepsilon_{ij}$ and $\mu_{ij}$ are dielectric constant and permeability of materials. $\beta_{ijk}$ and $\gamma_{ijk}$ are the second-order magneto-electric coupling coefficients. Therefore, the expressions of the polarization strength and the magnetization are:(40)$$P_{i}(\vec{E},\vec{H})=-\frac{\partial f}{\partial E_{i}}=P_{0i}+\varepsilon_{ij}E_{j}+\alpha_{ij}H_{j}+\frac{1}{2}\beta_{ijk}H_{j}H_{k}+...,$$
(41)$$M_{i}(\vec{E},\vec{H})=-\frac{\partial f}{\partial H_{i}}=M_{0i}+\mu_{ij}H_{j}+\alpha_{ji}E_{j}+\frac{1}{2}\gamma_{ijk}E_{j}E_{k}+....$$

According to Equations (40) and (41), $\alpha_{ij}$ is the first-order linear response of the electric polarization intensity (or magnetization intensity) to the magnetic field (or electric field), while $\beta_{jik}$ and $\gamma_{ijk}$ are the second-order responses.

In 2009, Giovannetti et al. used the DFT binding model method to predict the multiferroic of organic molecular crystal TTF-CA (tetrathiafulvalene-p-chloranyl) [12]. The two molecules are alternately arranged TTF-CA-TTF-CA forming a D–A array. TTF-CA shows a neutral-ionic transition at 81K, but there is small charge transfer (~0.3 e) even above the transition temperature. The entire array will undergo dimerization as shown in Figure 20a. At this time, the symmetry of the system is reduced, and the electric dipole moment appears as a whole. The electric polarization is approximately 3.5 µC∙cm^−2^. It was also found that the ground state of this system is antiferromagnetically coupled. Subsequently, Kagawa et al. studied the one-dimensional organic charge transfer system TTF-BA (tetrathiafulvalene-p-bromanil) and predicted that TTF-BA is almost ionic in a whole temperature region and undergoes a spin-Peierls transition at 53K. It does not show ferromagnetism but shows ferroelectricity [13]. 

TTF-CA is a charge transfer salt, TTF as an electron donor and CA as an electron acceptor. It is found that electrons on TTF spontaneously transition to CA and the ferroelectric phenomena and complete hysteresis loops were observed in many charge transfer salt systems. Spin-thermal electronics in organic materials are also quietly emerging, and this effect focuses on the interaction of spin and heat flow [78]. The thermoelectric and thermomagnetic effect has been applied to thermometers, generators and coolers. These studies include thermal conductivity and spin-dependent Seebeck/Peltier coefficients, thermal spin transfer torque and spin anomalous thermoelectric Hall effects.

In addition to the study of the charge transfer salt system, interesting phenomena have been found in the study of photoexcitation of organic compounds. In 2012, Ren et al. doped C_60_ in organic semiconducting single crystal poly-3(hexylthiophene) nanowires (nw-P3HT) [14]. By measuring the hysteresis loop of the sample, it is found that the maximum magnetic susceptibility of the sample is about 10 emu·cm^−3^ without light. At 615 nm and 20 mW of red light, the maximum magnetic susceptibility is raised to about 30 emu·cm^−3^ (as shown in Figure 21a). The experiment also found that both the electric field and the stress can regulate the magnetic susceptibility (as shown in Figure 21b), and no magnetic susceptibility was detected in both the nw-P3HT single crystal and the C_60_ elemental. It indicates that the charge transfer caused by illumination is the source of the system’s magnetic properties. This photoexcited ferromagnetism can also be regulated by an electric field, indicating that the material has the characteristics of magnetoelectric coupling. After that, photoexcited ferromagnetism was also observed in single-walled carbon nanotubes/ C_60_ system and nw-P3HT/Au system [15,16].

An organic composite chain composed of electron donor and acceptor can be described with Hamiltonian [79],

(42)$$H=H_{D}+H_{A}+H_{DA}.$$

In organic semiconductors, excitons are generated by photoexcitation of electrons from HOMO to LUMO. The probability of electronic transition is determined by the interband transition matrix element $\langle \psi_{LUMO}|H^{\prime }|\psi_{HOMO}\rangle$, where $H^{\prime }$ represents the photo-electrical interaction. In the non-relativistic approximation, the photo-electrical interaction mainly affects the spatial component of electrons. The spin remains conserved during the transition, and the photo-excited products of interband transition are mainly spin singlet excitons (SE). Considering the transition prohibition, the output of triplet excitons (TE) is negligible. However, if $H^{\prime }$ contains spin-related interactions, such as spin-flip effects, the yield of triplet excitons will increase significantly. If the electron is in a spin-mixed state, there is no pure state transition, and the SE and TE transitions evolve into EX1 and EX2 transitions (as shown in Figure 22b). For example, for excited EX1, the transition matrix element contains both spin conserved transitions and spin flip transitions, written as $\langle \psi_{LUMO}|H^{\prime }|\psi_{HOMO}\rangle =\langle \psi_{LUMO,s}|H^{\prime }|\psi_{HOMO,s}\rangle +\langle \psi_{LUMO,s}|H^{\prime }|\psi_{HOMO,-s}\rangle$. In organic charge transfer complexes, photoexcitation may occur within the molecules (donor or acceptor molecules) to form intramolecular excitons; it may also occur between molecules, where charge transfer occurs to form intermolecular excitons, also known as charge transfer state. Based on the above model, a calculation result is shown in Figure 23. For intramolecular excitons, the spin density distribution of the two excited states is shown in Figure 23a,c. It can be found that EX1 had a local spin density distribution with total net spin magnetic moment 1.98 *µ_B_*, where *µ_B_* is the Bohr magneton. EX2 had no spin. The spin density distributions of the intermolecular excitons EX1 and EX2 are shown in Figure 23b,d, respectively. For EX1, the calculation shows that the net magnetic moment on the electron donor was 0.91 *µ_B_* and the net magnetic moment on the acceptor was 0.96 *µ_B_*. Consider the same spin polarization, the total net magnetic moment was 1.87 *µ_B_*. For EX2, net magnetic moment on the electron donor was −0.85 *µ_B_* and the net magnetic moment on the acceptor was 0.93 *µ_B_*. Consider the opposite spin polarization direction, the total net magnetic moment was 0.08 *µ_B_*.

The spontaneous magnetization observed in the excited state is critical to understand the excited ferromagnetism in organic composites. Considering the spin-dependent interactions, the excited states of the system are intermolecular EX1 and EX2 states with spin mixing. Due to the intrinsic strong electron–phonon interaction of organic materials, they are spatially localized, similar to spin-quasiparticles, with local spin ${\vec{s}}_{1}$ and ${\vec{s}}_{2}$ respectively. As shown in Figure 24, $\left|{\vec{s}}_{1}\right|\neq \left|{\vec{s}}_{2}\right|$, regardless of whether ${\vec{s}}_{1}$ and ${\vec{s}}_{2}$ exhibit ferromagnetic coupling or anti-ferromagnetic coupling, the system always has a net magnetic moment. Experiments have also shown that, in nw-P3HT/C_60_, the coupling configuration of the two molecules has a certain effect on the ferromagnetic intensity. If the coupling configuration is similar to that shown in Figure 24, the system is likely to exhibit excited ferromagnetism. In fact, the configuration of Figure 24b is similar to the model of the organic ferromagnetic molecule poly-BIPO [80,81,82], and the spin coupling between EX1 and EX2 is described by the Heisenberg model $H_{H}=-J_{12}{\vec{s}}_{1}\cdot {\vec{s}}_{2}$. The coupling strength is related to the overlap integral or exciton density of the exciton state.

We noticed that one type of room-temperature chiral charge transfer magnet nw-P3HT/C_60_ was reported [83]. Compared to the achiral, it has not only good thermal stability but also good tunability on both saturation magnetization and magnetoelectric coupling effect under circularly polarized light. Yang et al. have found that organic charge transfer complex pyrene-F4TCNQ (pyrene-2,3,5,6-tetrafluoro-7,7,8,8-tetracyanoquinodimethane) can display anisotropic magnetoelectric coupling and changing the fluorine content in complexes, magnetoelectric coupling and magnetization can be tuned [84]. In particular, observation of the Cotton–Mouton effect in the complex provided a feasible path for the design of organic magnetoelectric devices.

## 5. Organic Spin Current

In a spin valve, the polarization current is injected into the organic layer from the ferromagnetic electrode by driving voltage. In 2013, Ando of Tohoku University and Watanabe of the Cambridge University discovered the existence of pure spin current in organic semiconductors [17,18]. They prepared device Ni_80_Fe_20_/PBTTT (poly(2,5-bis(3-alkylthiophen-2-yl)thieno[3,2-b]thiophene)/Pt. Structure of the device is shown in Figure 25.

By microwave excitation, spin is pumped to the organic polymer layer PBTTT. The spin current is converted into a current detected at the metal Pt end (inverse spin Hall effect, ISHE) due to spin-orbit coupling, which demonstrates the existence of the pure spin current. In this experiment, no external electric field was applied, thus there was no directional movement of carriers in the organic layer. The Hall bias measured in the Pt electrode is proportional to the spin current that reaches the electrode through the organic layer. The measurement of Hall voltage varies with the thickness of the organic layer is the exponential form, $V_{\mathrm{ISHE}}(d)\propto e^{-d/\lambda_{s}}$, $\lambda_{s}=153\pm 32\;\mathrm{nm}$ characterizes the spin decay length in the organic layer. The exponential behavior further demonstrates that the bias is generated by the ISHE in the non-magnetic Pt electrode. At the same time, the exponential behavior also indicates that the spin current in the organic layer is likely to be transported in some way. This study shows that pure spin currents can be achieved in organic semiconductors and are transported by spin polarons.

Meanwhile, Ando et al. directly measured the spin-charge conversion effects of PSS (poly (4-styrenesulphonate)) doped with the PEDOT (poly (3,4-ethylenedioxythiophene)) molecule [17]. They injected the spin current from the magnetic insulator Y_3_Fe_5_O_12_ vertically into the PEDOT: PSS by spin pumping. The conversion voltage measured in the gold electrodes on both sides is 600 nanovolts, which is two orders of magnitude smaller than the conversion voltage of metal Pt and close to the inorganic semiconductor silicon switching voltage (microvolt order of magnitude) [85]. The experimenter further found that, although the conversion bias is low, the organic spin-charge conversion efficiency is almost comparable to that of metal Pt. They believe that although the spin-orbit coupling in organic materials is much weaker than that of metal Pt, its longer spin relaxation time and conductivity anisotropy are conductive to improve the spin-charge conversion efficiency. Pulse-ferromagnetic resonance technique was used to inject spin current from ferromagnetic metal NiFe into organic semiconductors with different spin-orbit coupling strengths. The spin-charge conversion phenomena were also observed by the measurements of conversion bias and conversion efficiency [86]. Similarly, the spin-charge conversion phenomena in single-layer graphene have also been observed in spin-pumped experiments [87].

Yu proposed a polaron coupling model in organic semiconductors [88]. The Hamiltonian can be written as $H=H_{0}+H_{h}+H_{e}$, where:(43)$$H_{0}=\sum_{i}\varepsilon_{i}\left(a_{i↑}^{+}a_{i↑}+a_{i↓}^{+}a_{i↓}\right)-g\mu_{B}{\vec{s}}_{i}\cdot \vec{B},$$
(44)$$H_{h}=\sum_{\langle ij\rangle s}V_{ij}\left(a_{is}^{+}a_{js}+a_{js}^{+}a_{is}\right),$$
(45)$$H_{e}=\sum_{ij}J_{ij}{\vec{s}}_{i}\cdot {\vec{s}}_{j},$$
where $a_{is}^{+}$ indicates that a polaron with spin *s* is generated on molecule *i*. $\varepsilon_{i}$ is the positional energy of the polaron. Spin of the polaron can be written as ${\vec{s}}_{i}=\left(\hbar /2\right)\sum_{\sigma \sigma^{\prime }}a_{i\sigma }^{+}{\vec{\sigma }}_{\sigma \sigma^{\prime }}a_{i\sigma^{\prime }}$. $V_{ij}$ is the polaron transition integral, and $J_{ij}$ is the exchange integral between polarons. 

If we considered the incoherent transport, polaron hopping between molecules is described by the master equation, and the spin polarization at molecule *i* follows the kinetic equation,
(46)$$\frac{d{\vec{M}}_{i}}{dt}={\vec{M}}_{i}\times \vec{\omega }-\sum_{j}w_{ij}(1-f_{j})\left({\vec{M}}_{i}-{\vec{M}}_{j}\right)-\sum_{j}f_{j}\eta_{ij}\left({\vec{M}}_{i}-{\vec{M}}_{j}\right),$$
where $\vec{\omega }=\frac{e}{2m}\vec{B}$, $f_{j}$ is the probability of a polaron at molecule *j* and determined by the master equation,
(47)$$\frac{df_{j}}{dt}=-\sum_{i}\left[w_{ji}f_{j}(1-f_{i})-w_{ij}f_{i}(1-f_{j})\right],$$
where $w_{ij}$ indicates the hopping rate of the polaron from molecule *i* to *j* per unit time, which can be given by the Marcus transition formula. $\eta_{ij}=\sqrt{\pi }J_{ij}^{2}/\omega_{e}$, $\omega_{e}^{2}=\sum_{j}8J_{ij}^{2}S(S+1)/3≃12{\bar{J}}^{2}$. Polaron hopping occurs between occupied and unoccupied molecules, as described by the second term on the right side of Equation (46); spin exchange of polaron occurs between occupied molecules and is given by the third item on the right of Equation (46), which contributes to the spin distribution.

When the spin polarization changes slowly in space, the kinetic equation can be reduced to the continuous equation:(48)$$\frac{d\vec{M}}{dt}=\left(D_{h}+D_{e}\right)\nabla^{2}\vec{M}+\vec{M}\times \vec{\omega },$$
where $D_{h}=\bar{w}{\bar{a}}^{2}$ and $D_{e}=\bar{\eta }{\bar{R}}^{2}\equiv \sqrt{\pi /12}\bar{J}\;{\bar{R}}^{2}$ are hopping-induced and exchange-induced spin diffusion (SD) constants respectively. $\bar{w}(\bar{\eta })$ is the ensemble average of $w_{ij}(\eta_{ij})$, and $\bar{a}(\bar{R})$ is the average spacing of molecules (polarons). Since $D_{e}=0$, the diffusion constant of charge and spin is the same in inorganic materials. $D_{h}=0$ in magnetic insulators, so spin transport is mainly generated by exchange. The spin-exchange coupling between polarons is determined by their interaction and the overlap of wave functions. Since the polaron is a localized state $\phi \sim e^{-r/\xi }$, the exchange coupling has the following form:(49)$$\bar{J}=0.821\frac{e^{2}}{\varepsilon \xi }{\left(\frac{\bar{R}}{\xi }\right)}^{5/2}e^{-2\bar{R}/\xi },$$
where ε is the dielectric constant and ξ is the local length of the polaron state. The average distance of the polarons is related to its concentration, $\bar{R}=n^{-1/3}$, $n=\sum_{i}f_{i}/V$. For the Alq_3_ molecule, ε=2, ξ=1 and fixed $D_{h}$. Figure 26 gives the relationship between exchange-induced spin diffusion and polaron concentration. It can be seen that when the polaron concentration n exceeds 10^17^ cm^−3^, the exchange-induced spin diffusion begins to become apparent and quickly dominates.

## 6. Organic Spin Device Outlook

Since the 1980s, the major progress or breakthrough comes about every decade. In the 1980s, the focus of research was on the conductivity of organic materials, which is throughout the research of organic semiconductors; in the 1990s, organic light-emitting achieved a major breakthrough. At present, the organic display has gone to market, and optimized research is still going on. Organic spintronics has been the focus of research for the past decade. New phenomena such as organic spin valves, organic strong magnetic effects, organic multiferroics and organic spin currents are emerging one after another.

Organic semiconductors have abundant properties in electromagnetism and optics, which can be used to prepare various electronic devices, such as OLEDs, organic transistors and organic solar cells. Monochromatic and multi-color displays based on OLEDs have entered the commercial application stage. As new phenomena continue to be revealed, some novel organic devices may be fabricated. For example, OLED luminescence is mainly based on spin singlet excitons, which can be controlled by a magnetic electrode or the external magnetic field to realize the magnetic regulation of luminescence.

Some studies indicated that considerable spin-orbit coupling strength of organic materials could be achieved in some cases. First, some molecules contain heavy atoms in their own components. For example, the current Alq_3_ molecule wildly studied contains the Al element, and the CuPc molecule contains the Cu element. When the molecule contains S or an atom with a larger atomic number, its spin-orbit coupling can have a significant effect on spin relaxation. Theoretical calculation shows the dependence of organic spin diffusion length on temperature and electric field, indicating that the Elliott–Yafet mechanism caused by spin-orbit coupling plays an important role in organic spin relaxation [89]. This also makes the introduction of heavy elements an organic one of the effective ways to artificially design organic materials with strong spin-orbit coupling. For example, Liu et al, from the University of Utah, have theoretically designed a two-dimensional organometallic mesh by introducing heavy metal elements (Mn, In, Ti, etc.) into the triphenyl molecules, and the first-principles calculation shows that this structure can realize a topological insulator, quantum anomalous hall effect and other phenomena [90]. On the other hand, studies have shown that changes of organic molecular structure and orbital hybridization can also result in enhancement of spin-orbit coupling strength even without heavy elements. Such as Shuai et al. DFT study shows that the distortion between carbon ball substituents or benzene rings changes the spin-orbit coupling, and then changes the intersystem transition [91]; according to the theoretical calculations, Yu pointed that when the two biphenyl rings of biphenyl molecules are perpendicular to each other rather than coplanar structures, spin-orbit coupling strength of the system can be increased by nearly four orders of magnitude [92]. It is worth mentioning that when the orbital hybridization of individual carbon atoms changes from *sp*^2^ to *sp*^3^ by introducing structural defects to a single-layer graphene containing only carbon atoms, the spin-orbit coupling is improved by nearly three orders of magnitude [93]. In terms of order of magnitude of spin-orbit couplings strength, bending carbon nanotubes is comparable to GaAs semiconductors, which results from variation of *sp*^2^ hybrid state caused by the curved structure. The peculiar phenomenon has aroused great research interest in the past few years. Structural diversity of organic molecules and abnormally abundant methods of structural modulation, such as changing the torsion angle, side group substitution, tensile compression, etc., provide many feasible paths to enhance spin-orbit coupling in pure organic materials.

In 2006, the Naaman group, in Israel, adsorbed a double-stranded DNA monolayer on the Au substrate. It is found that its photoelectron transmission spectrum has a spin polarization phenomenon, which is called CISS (chiral-induced spin selectivity) [94]. The effect does not seem to exist in a single helix DNA molecule [95]. However, spin polarization of transmitted electrons in bacteriorhodopsin (a single *α*-helix bacterial rhodopsin) was found by Mishra et al. in 2013 and Einati et al. in 2015, respectively [96,97]. Therefore, the CISS effect should be a general feature of chiral molecules or chiral nanostructured materials.

Spiral or axial chirality is an important type of chiral structure. DNA and proteins have a helical chiral structure, and carbon nanotubes can also form chiral structures. Chiral molecules can self-assemble into nanostructures, superstructures and even macrostructures, and their unique structural spin-orbit coupling can produce many novel spin-charge correlation phenomena. “Chiral Electronics” or “Chiral Spintronics” has quietly appeared [98,99,100,101,102,103,104]. Researches on organic chiral spin-related phenomena not only promote the development of organic spintronics, promote the intersection of physics and chemistry and micro-nano electronics and develop the application of organic chiral molecules and fibers in functions such as electromagnetism optics and others; but also contribute to understand the information storage and transmission of biological macromolecules, provide a physical perspective of the complex life and design more organic (spin) electronics that are stretchable, wearable or implantable.

Over the past few years, more and more functional properties of organic materials have been exploited, including electrical, magnetic and optical properties. We believed that all-organic devices would appear in the future. They have functions of display, sensing and recognition, and have features such as low energy consumption, low cost and foldability. This chapter reviews the recent developments in organic spintronics, including the theoretical work of our group related to the chapter. Organic materials are soft, resourceful and inexpensive. Compared to inorganic materials, organic functional materials have advantages in the application fields of future electronic and spintronic devices. All these properties motivate people to develop organic functional devices. The development of this field is not so smooth, such as the instability of experimental data, the diversity of theoretical models and so on. Nevertheless, research on organic spintronics has only just begun, people’s enthusiasm for organic spintronics is growing. We believed that persistent research on organic functional materials and organic devices will contribute to understanding the relevant effect mechanisms that are not yet clear and explore better application potential.

## Figures and Tables

**Figure 1 micromachines-10-00596-f001:**
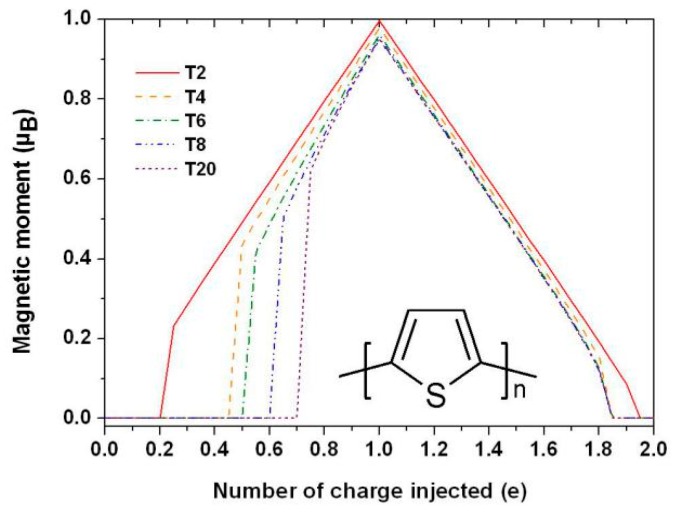
Magnetic moment varies with the quantity of charges injected in different thiophene aggregation degrees. The inset shows the unit structure of the thiophene molecule. By Han et al. [23] with the permission of Organic Electronics.

**Figure 2 micromachines-10-00596-f002:**
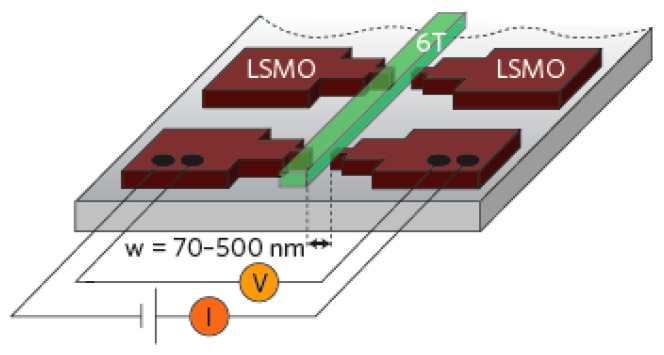
Structure diagram of the LSMO/T6/LSMO sandwich device.

**Figure 3 micromachines-10-00596-f003:**
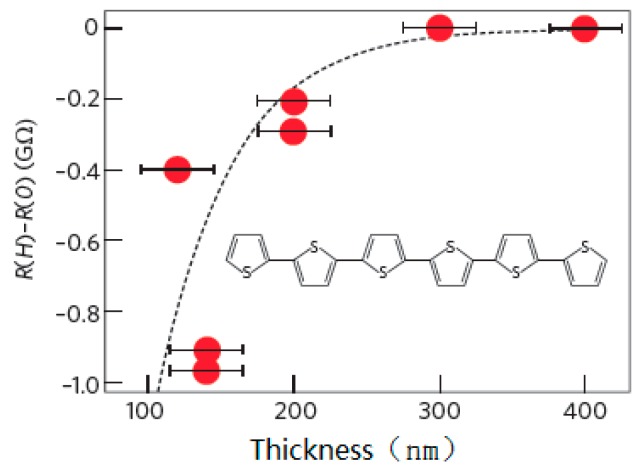
Relation between magnetoresistance (MR) and T6 thickness.

**Figure 4 micromachines-10-00596-f004:**
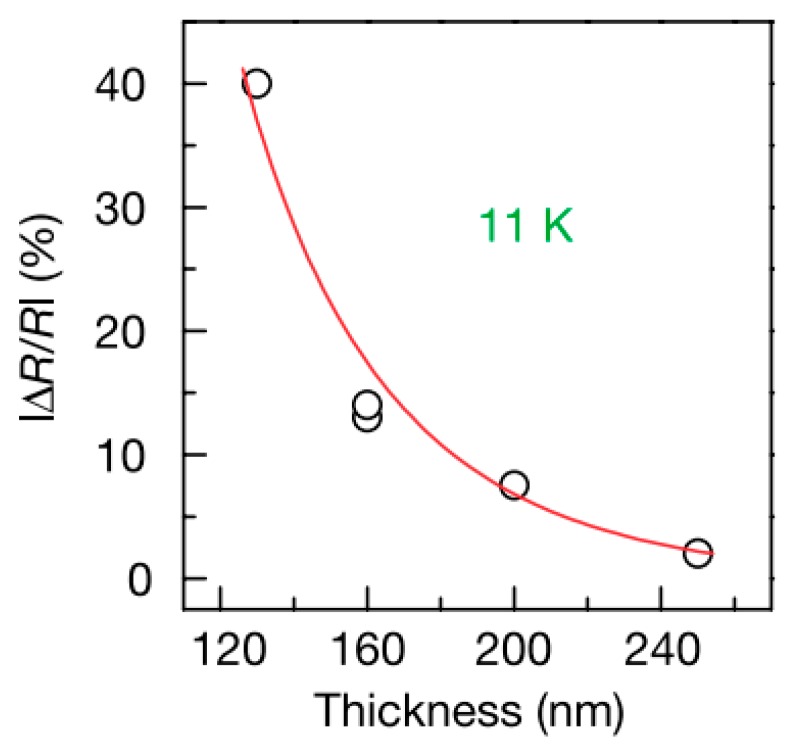
MR changes with the thickness of the organic layer by Xiong et al. [9] with the permission of Nature.

**Figure 5 micromachines-10-00596-f005:**
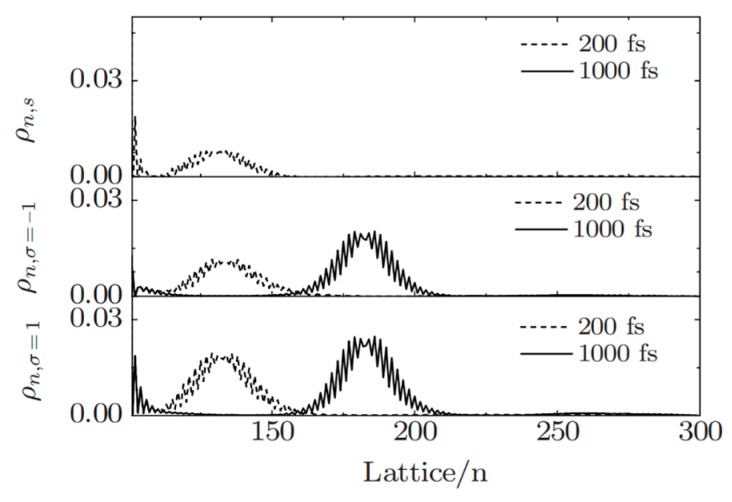
The distribution of spin up, spin down and net spin density in organic layer at different times, where the bias voltage was *V* = 0.85 eV and the electric field was *E* = 0.5 mV/nm. By Fu et al. [39] with the permission of Physical Review B.

**Figure 6 micromachines-10-00596-f006:**
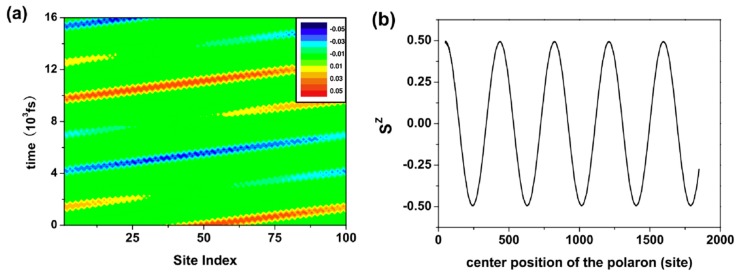
(**a**) Spin evolution of polarons, positive values denote spin up, negative values denote spin down and zero denote spinless; and (**b**) the polaron spin varies with the position of the center. By Lei et al. [33] with the permission of the Journal of Physics: Condensed Matter.

**Figure 7 micromachines-10-00596-f007:**
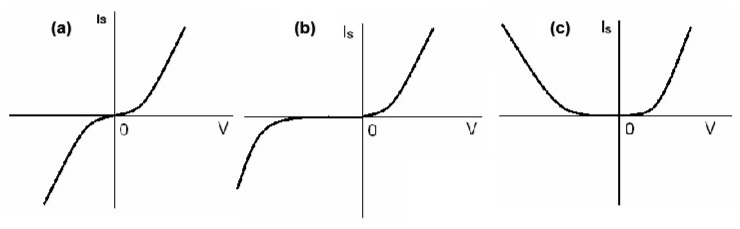
Spin rectification: (**a**) Symmetrical spin rectification; (**b**) parallel spin rectification and (**c**) anti-parallel spin current rectification.

**Figure 8 micromachines-10-00596-f008:**
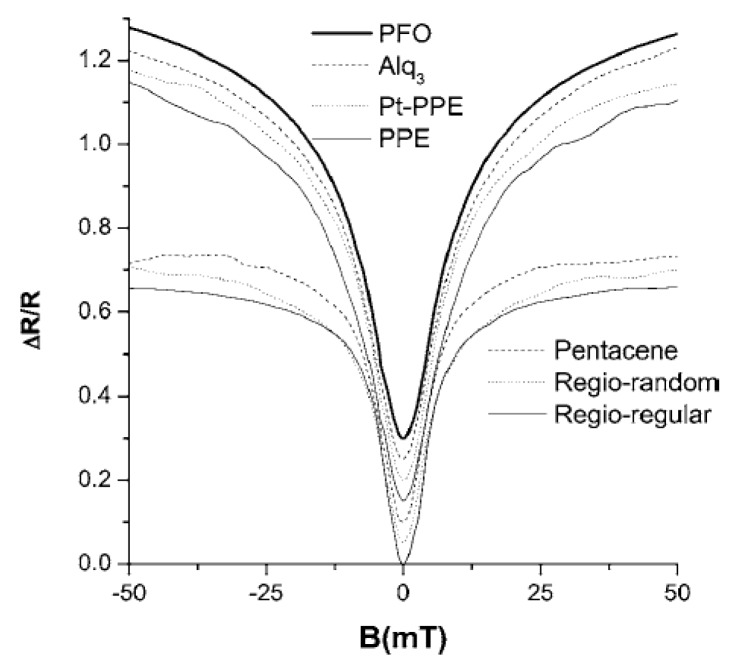
The fitting curve of the MR of different organic materials with the change of magnetic field by Mermer et al. [11] with the permission of Physical Review B.

**Figure 9 micromachines-10-00596-f009:**
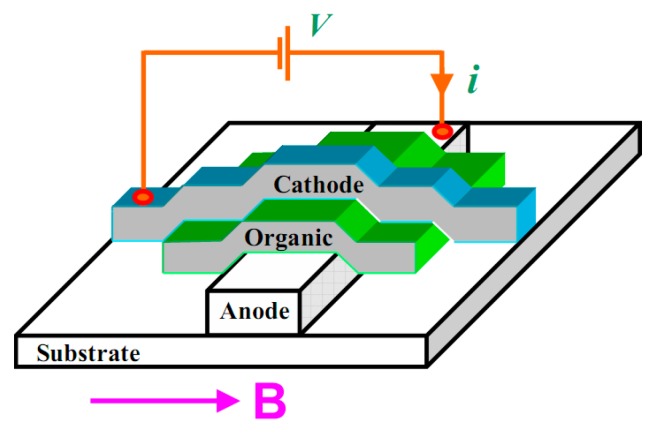
Schematic diagram of the experimental device corresponding to the magnetic field effect of organic devices by Mermer et al. [11] with the permission of Physical Review B.

**Figure 10 micromachines-10-00596-f010:**
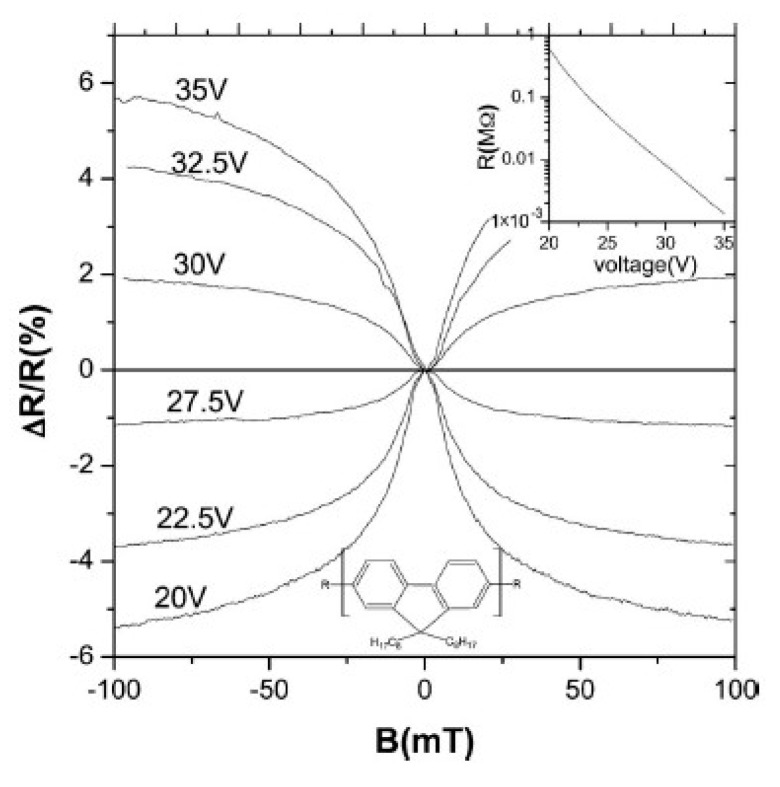
MR versus magnetic field in ITO/PFO (≈ 60 nm)/Ca devices. The inset shows the variation of device resistance with voltage by Mermer et al. [11] with the permission of Physical Review B.

**Figure 11 micromachines-10-00596-f011:**
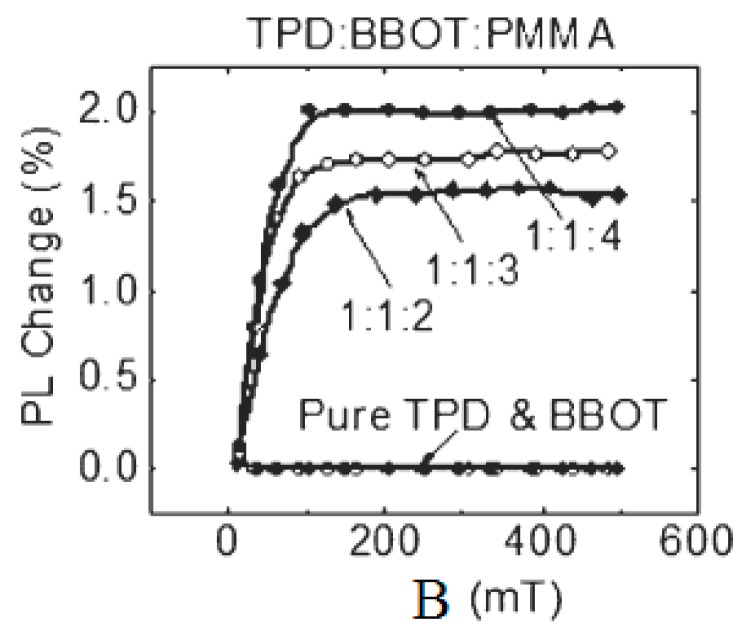
Photoluminescence efficiency of organic mixtures with different ratios varies with the magnetic field by Shao et al. [49] with the permission of Advanced Materials.

**Figure 12 micromachines-10-00596-f012:**
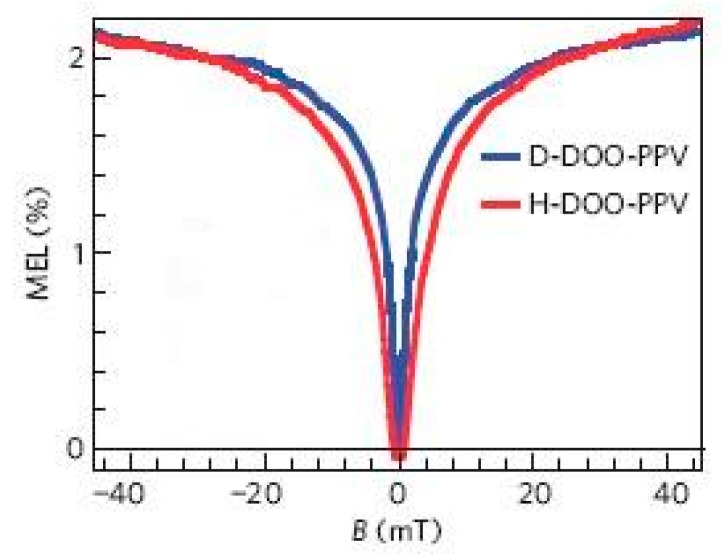
Effect of isotope effect on magnetoluminescence efficiency in organic light diodes (OLEDs) by Nguyen et al. [32] with the permission of Nature Materials.

**Figure 13 micromachines-10-00596-f013:**
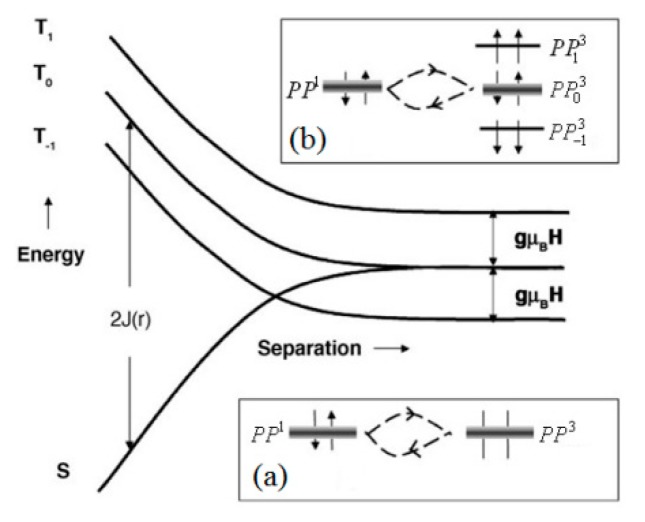
Structure change of single and triplet energy levels with different distances. The insets are schematic diagrams of the mutual transformation between singlet and triplet polaron pairs when (**a**) *B* = 0 and (**b**) *B* ≠ 0. By Prigodin et al. [61] with the permission of Synthetic Metals.

**Figure 14 micromachines-10-00596-f014:**
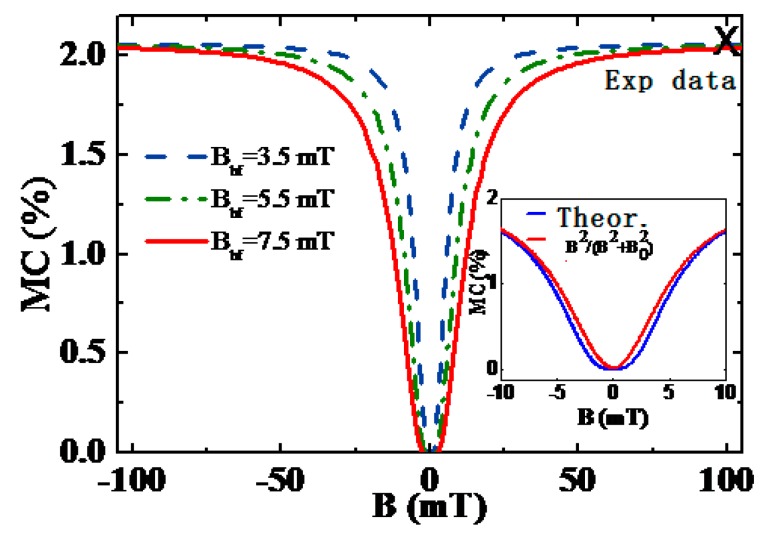
MC curve with the magnetic field in the case of different hyperfine equivalent fields, *B_hf_* = 3.5, 5.5 and 7.5 mT. Symbol × is the experimental value at *B* = 100 mT. By Qin et al. [34] with the permission of Applied Physics Letters.

**Figure 15 micromachines-10-00596-f015:**
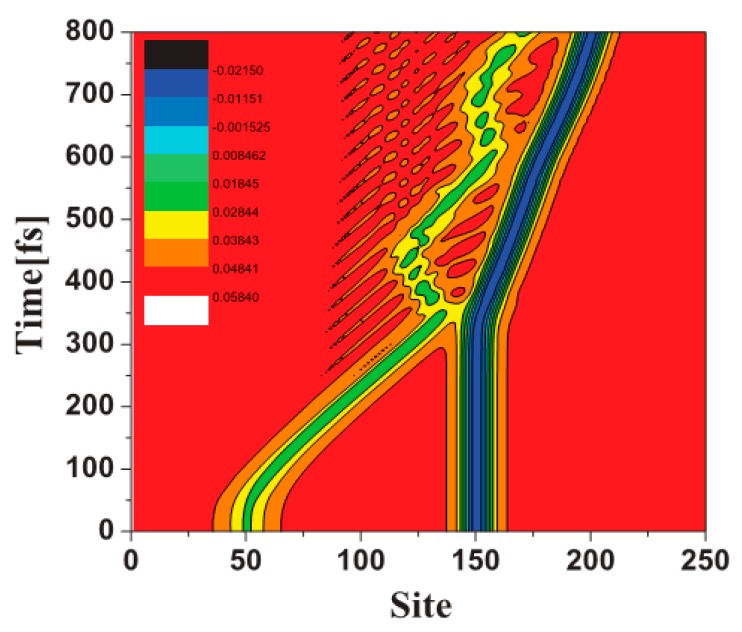
The evolution of polaron and triplet exciton lattice configuration under electric field with time by Su et al. [64] with the permission of Journal of Chemical Physics.

**Figure 16 micromachines-10-00596-f016:**
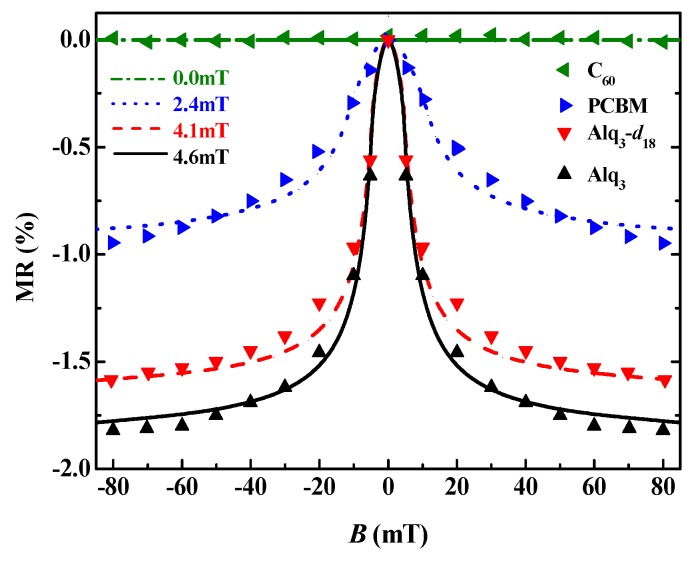
Dependence of MR with the magnetic field under a different hyperfine field. ▼ represents the experimental data [66,67]. The curves represent the theoretical results [65].

**Figure 17 micromachines-10-00596-f017:**
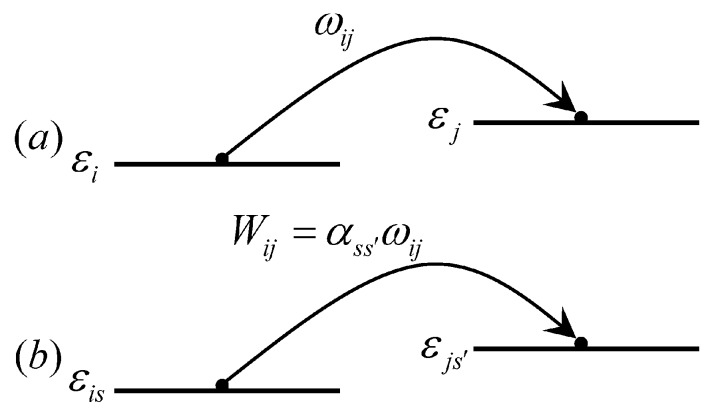
Schematic diagram of the polaron transition: (**a**) A transition without a spin state and (**b**) a transition containing a spin state.

**Figure 18 micromachines-10-00596-f018:**
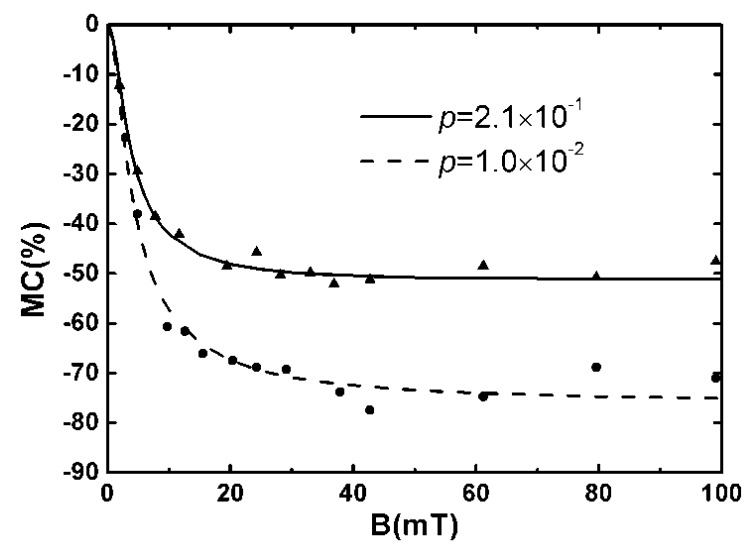
Dependence of the magnetic conductance on the magnetic field at different polaron concentrations by Yang et al. [68] with the permission of the Journal of Chemical Physics. Triangles and dots are experimental data [69].

**Figure 19 micromachines-10-00596-f019:**
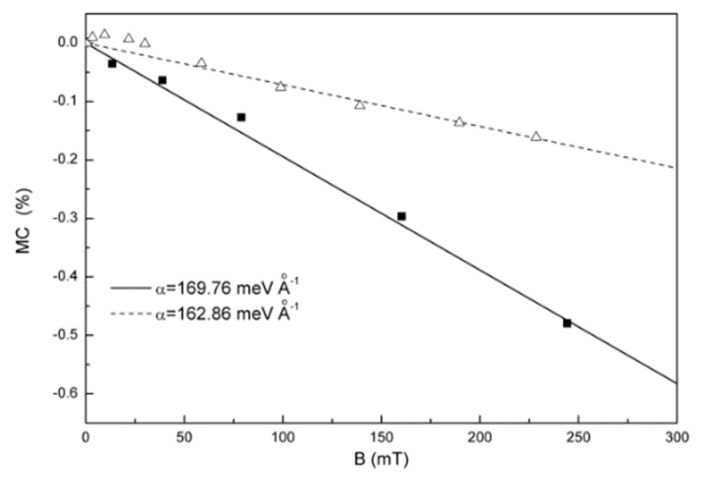
Change of magnetic conductance under different electron phonon coupling. Lines are the theoretical results by Li et al. [71] with the permission of Organic Electronics. Data for the block symbols are for PtOEP [72], and triangle symbols for Ir(ppy)_3_ [73].

**Figure 20 micromachines-10-00596-f020:**
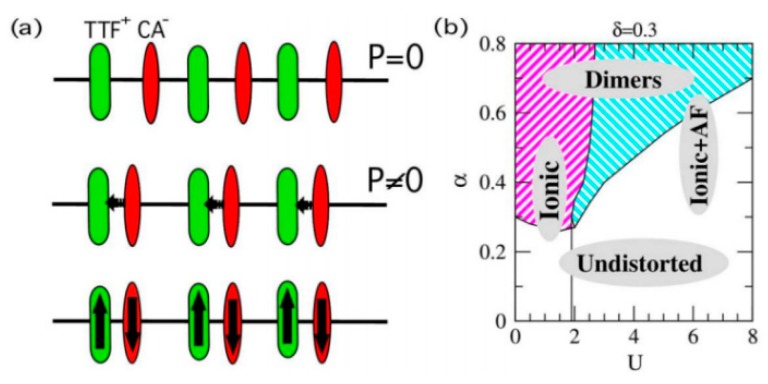
(**a**) Schematic diagram of the TTF-CA molecular crystal. The figure above shows the molecular arrangement diagram when the distribution is uniform, and the electrical polarization of the system is zero. The middle diagram is the molecular arrangement diagram after dimerization, and the system is electrically polarized. The figure below shows the system antiferromagnetic coupling and the (**b**) phase diagram of the system magnetism varying with electron–phonon coupling and electron–electron interaction. By Giovannetti et al. [12] with the permission of Physical Review Letters.

**Figure 21 micromachines-10-00596-f021:**
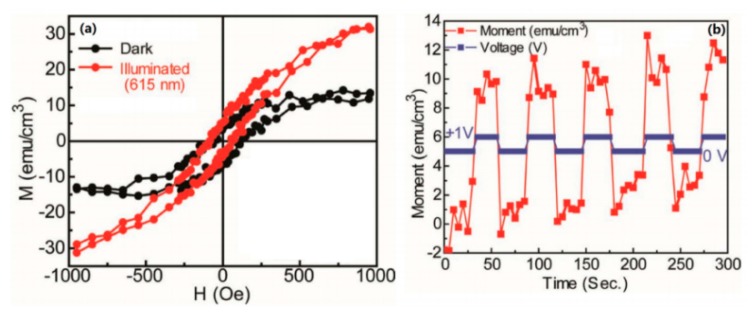
(**a**) In the nw-P3HT/C_60_ device, the hysteresis loop of the device before and after illumination and (**b**) the magnetization of the nw-P3HT/C_60_ device with the bias voltage, indicating that the device has magnetoelectric coupling nature. By Ren et al. [14] with the permission of Advanced Materials.

**Figure 22 micromachines-10-00596-f022:**
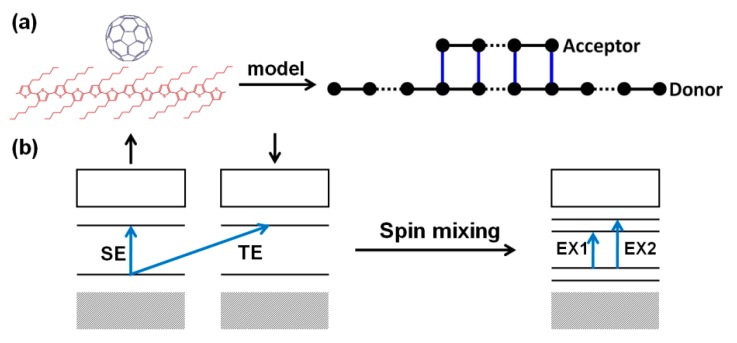
(**a**) Schematic diagram of the nw-P3HT/C_60_ charge transfer complex device and corresponding theoretical modeling and (**b**) transitions of photoexcited electrons between HOMOand LUMO before and after spin mixing.

**Figure 23 micromachines-10-00596-f023:**
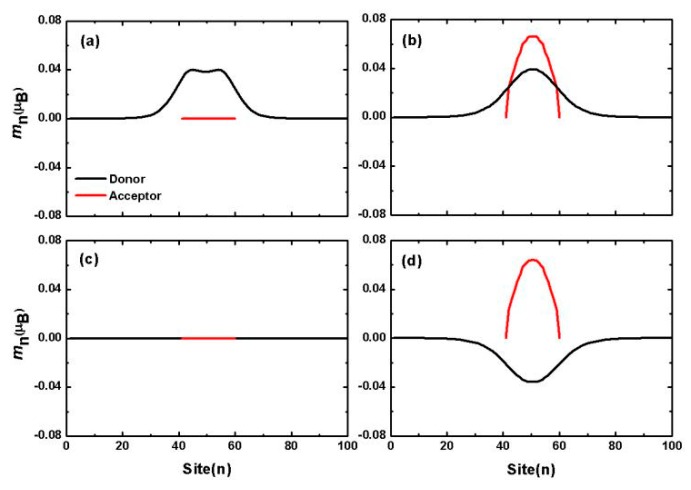
Intramolecular (**a**,**c**) and intermolecular (**b**,**d**) exciton spin density distribution. (**a**,**b**) For exciton EX1 and (**c**,**d**) for exciton EX2.

**Figure 24 micromachines-10-00596-f024:**
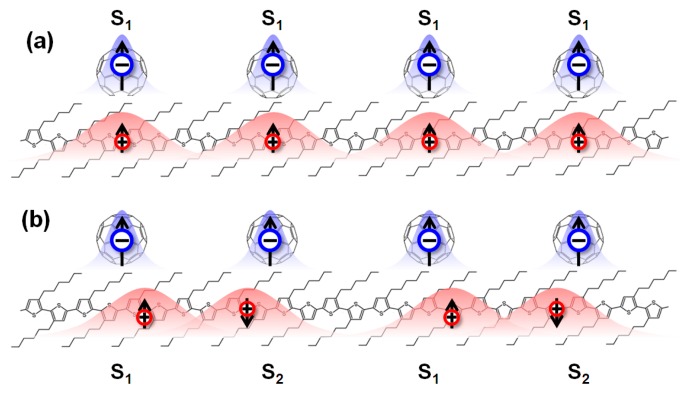
Schematic diagrams of (**a**) only triplet or EX1 excitons in parallel configuration, and (**b**) alternative EX1 and EX2 configuration.

**Figure 25 micromachines-10-00596-f025:**
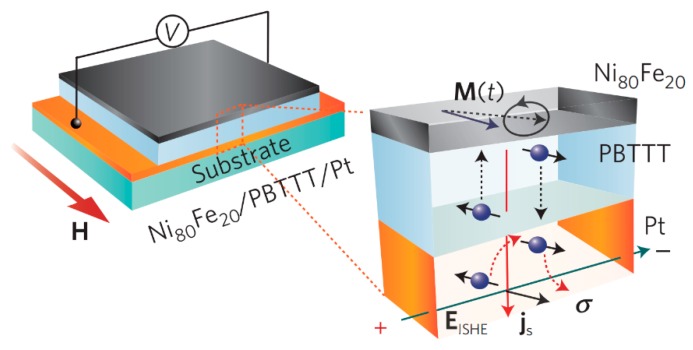
Schematic diagram of an organic pure spin current device by Watanabe et al. [18] with the permission of Nature Physics. The middle layer is an organic polymer film.

**Figure 26 micromachines-10-00596-f026:**
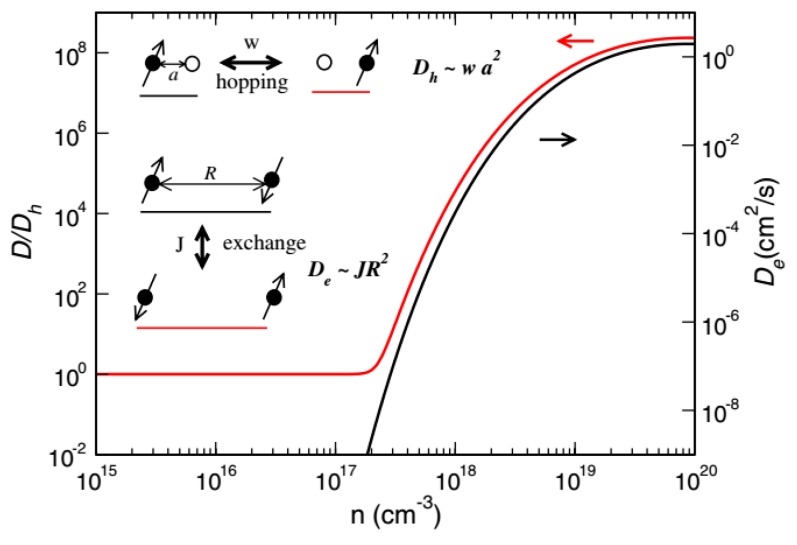
Relationship between the spin diffusion (SD) constant and polaron density by Yu et al. [88] with the permission of Physical Review Letters. Black and red (gray) lines are $D_{e}$ and $D/D_{h}$ respectively, where $D_{h}=\nu_{h}k_{B}T/e$ and $\nu_{h}=10^{-6}cm^{2}/s$. The inset illustrates hopping-induced and exchange-induced and solid (open) circles represent occupied (vacant) sites.

**Table 1 micromachines-10-00596-t001:** Charge and spin relationship of organic carriers.

Carrier	Charge (e)	Spin (ħ)	Degeneracy
Neutral soliton	0	1/2	Degenerate
Charged soliton	±1	0	Degenerate
Polaron	±1	1/2	Degenerate, Non-degenerate
Bipolaron	±2	0	Non-degenerate
Singlet exciton	0	0	Non-degenerate
Triplet exciton	0	1	Non-degenerate
Biexciton	0	0	Non-degenerate
Trion	±1	1/2	Non-degenerate

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
