# Peer review of "Spin Injection and Transport in Organic Materials"

_micromachines, 2019, doi:10.3390/mi10090596_

Round 1
Reviewer 1 Report
This paper reviews recent progress of spin injection and transport in organic materials as well as some important discoveries. The authors mention some interesting phenomena such as low field magnetoresistance and introduce theoretical explanations.
However, it would be better if they include more basic issue like conductance mismatch and discuss how to introduce spin polarized current efficiently.
The paper includes many figures from other original papers. However, there is no mention about copyright. Please check it.
There are many undefined words for chemicals: HO-PPV, BIPO, PFO, TPD, BBOT, PMMA, DOC-PPV, NPB, TTF, CA, BA, P3HT.
There is no definition for D in eq. 5.
The line 127: There is no reference in “Tsukagoshi et al prepared….”
The lines 215-216: “The figure also shows that the injected charge still has obvious spin polarization after 1000fs.” It is not consistent with the figure 5 because there is no data after 1000fs.
Figure 6 (a) is not explained in the text. What does the color mean in Fig. 6 (a)?
The line 437: What is the OMFE?
The lines 472-476: These sentences are wrong.
TTF–CA shows a Neutral-Ionic transition at 81 K, but there is small charge transfer (~ 0.3e) even above the transition temperature. TTF–BA is almost ionic in a whole temperature region and undergoes a spin-Peierls transition at 53 K. It does not show ferromagnetism but shows ferroelectricity.
There are many references, but very recent one is very limited: Only a few for chiral system [91-96]. Therefore, the reviewer wonders if very recent papers are considered appropriately.
Finally, the authors should follow the style guides because of confusion in reference.
Author Response
Thanks for the reviewer's report on the "Spin Injection and Transport in Organic Materials" with the Manuscript ID of micromachines 560066. We highly appreciate the comments from reviewer, by which we have carefully revised the manuscript. The “Point-to-point response” to the reviewer’s comments are listed as follows. We sincerely hope that this response as well as the revised manuscript will meet the requirements.
Point 1: It would be better if they include more basic issue like conductance mismatch and discuss how to introduce spin polarized current efficiently.
Response 1: We have added some related contents on page 6-7 in the revised manuscript.
Point 2: The paper includes many figures from other original papers. However, there is no mention about copyright. Please check it.
Response 2: Thanks for your reminding. We have applied for and obtained the copyright of all the pictures in the article, which will be attached to this submission
Point 3: There are many undefined words for chemicals: HO-PPV, BIPO, PFO, TPD, BBOT, PMMA, DOC-PPV, NPB, TTF, CA, BA, P3HT.
Response 3: In response to your suggestion, we have explained the words mentioned in detail in the revised manuscript, as follows :
HO-PPV=poly(2,5-diheptyloxy-p-phenylene vinylene),
BIPO=poly (1,4-bis(2,2,6,6-tetramethyl-4-piperidyl-1-oxyl) -butadiin,
PFO=poly(9,9-dioctylfluorenyl-2,7-diyl),
TPD=N,N’-diphenyl-N,N’-bis(3-methylphenyl)-[1,1’-biphenyl]-4,4’-diamine,
BBOT=2,5-bis(5-tert-butyl-2-benzoxazolyl)-thiophene,
PMMA=poly(methyl methacrylate),
DOO-PPV=poly(dioctyloxy)phenylenevinylene,
NPB=N,N’-bis(l-naphthyl)-N,N’-diphenyl-1,1’-biphentl-4,4’-diamine,
TTF-CA=tetrathialfulvalene-p-chloranyl,
TTF-BA=tetrathiafulvalene-p-bromanil,
nw-P3HT=poly-3(hexylthiophene) nanowires.
Point 4: There is no definition for D in eq. 5.
Response 4: We have added the definition of in the revised manuscript (p.5).
Point 5: The line 127: There is no reference in “Tsukagoshi et al prepared….”
Response 5: Sorry for our carelessness. We have added the reference as ref.[27] in the revised manuscript.
Point 6: The lines 215-216: “The figure also shows that the injected charge still has obvious spin polarization after 1000fs.” It is not consistent with the figure 5 because there is no data after 1000fs.
Response 6: Sorry for our carelessness and thank you for your careful checking. We indeed gave a wrong figure in the last manuscript. We have given the correct figure in the revised manuscript (p.8)
Point 7: Figure 6 (a) is not explained in the text. What does the color mean in Fig. 6 (a)?
Response 7: We have added legend and explained the meaning of color in the revised manuscript.
Point 8: The line 437: What is the OMFE?
Response 8: We have explained the abbreviation OMFE ( Organic Magnetic Field Effect ) in the revised manuscript.
Point 9: The lines 472-476: These sentences are wrong.
Response 9: Many thanks! We have rewritten this paragraph referring to the referee’s suggestion.
Point 10: There are many references, but very recent one is very limited: Only a few for chiral system [91-96]. Therefore, the reviewer wonders if very recent papers are considered appropriately.
Response 10: Thanks for your comments. We have researched the recent progress on organic spintronics and added 15 references published in the last 3 years in the revised manuscript.
Point 11: Finally, the authors should follow the style guides because of confusion in reference.
Response 11: We have reformatted the references in the revised manuscript.

Reviewer 2 Report
This review article written by Tian and Xie is discussing the development of spintronics, both the experimental work and theoretical work. The architecture of the article is fine and the discussions are also good for publication. There are a few minor revisions needed and the language should be polished a little more before publication.
Line 173, Landau-Buttiker theory; Line 240 Landauer-buttiker formula
Line 247: “MR follows Lorentz law……” the expressions here are not even formulas. The authors should correct this.
Line 391: “the above-described coherent……” should be changed into “ the coherent transport mechanism described before is not suitable”
Line 489, this sentence needs rewriting. It’s not clear what the authors mean here.
Figure 26, the figure need more description. What are the axis. The left and right y axis.
Author Response
Thanks for the reviewer's report on the "Spin Injection and Transport in Organic Materials" with the Manuscript ID of micromachines 560066. We highly appreciate the comments from reviewer, by which we have carefully revised the manuscript. The “Point-to-point response” to the reviewer’s comments are listed as follows. We sincerely hope that this response as well as the revised manuscript will meet the requirements.
Point 1: Line 173, Landau-Buttiker theory; Line 240 Landauer-buttiker formula
Response 1: Thanks for pointing out, we have unified the terms in revised manuscript.
Point 2: Line 247: “MR follows Lorentz law……” the expressions here are not even formulas. The authors should correct this.
Response 2: Thanks for your seriousness, we have rewritten this sentence in revised manuscript
Point 3: Line 391: “the above-described coherent……” should be changed into “ the coherent transport mechanism described before is not suitable”
Response 3: Many thanks! We have rewritten this sentence referring to the referee’s suggestion.
Point 4: Line 489, this sentence needs rewriting. It’s not clear what the authors mean here.
Response 4: We have rewritten that sentence and added more description to make it clear.
Point 5: Figure 26, the figure need more description. What are the axis. The left and right y axis.
Response 5: Sorry for our carelessness and we have added more descriptions in the figure and defined the axes in revised manuscript.
Reviewer 3 Report
This review paper tries to cover far too much research on organic spintronics and related topics such that it is a series of unrelated paragraphs that do not follow very well. A lot of equations are presented but not all the symbols are defined and often they are used for different meaning in the paper, this is confusing for the reader.
The paper is too broad and doesn't really go into enough detail on any of the subjects it covers, a better review would be to cover one of the areas and make it more detailed allowing focus on the meaning the equations rather than just putting them in without much understanding.
Author Response
Thanks for the reviewer's report on the "Spin Injection and Transport in Organic Materials" with the Manuscript ID of micromachines 560066. We highly appreciate the comments from reviewer, by which we have carefully revised the manuscript. The “Point-to-point response” to the reviewer’s comments are listed as follows. We sincerely hope that this response as well as the revised manuscript will meet the requirements.
Point 1: This review paper tries to cover far too much research on organic spintronics and related topics such that it is a series of unrelated paragraphs that do not follow very well. A lot of equations are presented but not all the symbols are defined and often they are used for different meaning in the paper, this is confusing for the reader.
Response 1: Thanks for your suggestion. Organic spintronics is a rapid developing area. We try to do our best to review the progress on spin injection, transport and magnetic field effect. We focus it more in physics than technology in present manuscript. Off course, limited by our knowledge, we are difficult to give a systematic and profound review.
We have polished the whole manuscript including symbols, abbreviations and sentences. We hope that the revised manuscript is much better than the last one in reading and understanding.
Point 2: The paper is too broad and doesn't really go into enough detail on any of the subjects it covers, a better review would be to cover one of the areas and make it more detailed allowing focus on the meaning the equations rather than just putting them in without much understanding.
Response 2: Thanks for the reviewer's comments and we agree with this very much. We also hope that our review can cover more details. However, due to the limitation of the length of the article, we have to reduce the content, which may cause a lack of close connection between the subjects. Anyway, we have polished the whole manuscript referring to all the reviewers’ suggestions.